# Adaptive Federated Learning via Dynamical System Model

**Aayushya Agarwal**                                  *aayushya@andrew.cmu.edu*
*Department of Electrical and Computer Engineering*
*Carnegie Mellon University*
**Larry Pileggi**                                     *pileggi@andrew.cmu.edu*
*Department of Electrical and Computer Engineering*
*Carnegie Mellon University*
**Gauri Joshi**                                       *gaurij@andrew.cmu.edu*
*Department of Electrical and Computer Engineering*
*Carnegie Mellon University*

**Reviewed on OpenReview:** *https://openreview.net/forum?id=ebDfXAklXg*

## Abstract

Hyperparameter selection is critical for stable and efficient convergence of heterogeneous federated learning, where clients differ in computational capabilities, and data distributions are non-IID. Tuning hyperparameters is a manual and computationally expensive process as the hyperparameter space grows combinatorially with the number of clients. To address this, we introduce an end-to-end adaptive federated learning method in which both clients and central agents adaptively select their local learning rates and momentum parameters. Our approach models federated learning as a dynamical system, allowing us to draw on principles from numerical simulation and physical design. Through this perspective, selecting momentum parameters equates to critically damping the system for fast, stable convergence, while learning rates for clients and central servers are adaptively selected to satisfy accuracy properties from numerical simulation. The result is an adaptive, momentum-based federated learning algorithm in which the learning rates for clients and servers are dynamically adjusted and controlled by a single, global hyperparameter. By designing a fully integrated solution for both adaptive client updates and central agent aggregation, our method is capable of handling key challenges of heterogeneous federated learning, including objective inconsistency and client drift. Importantly, our approach achieves fast convergence while being insensitive to the choice of the global hyperparameter, making it well-suited for rapid prototyping and scalable deployment. Compared to state-of-the-art adaptive methods, our framework is shown to deliver superior convergence for heterogeneous federated learning while eliminating the need for hyperparameter tuning both client and server updates.

## 1 Introduction

Federated learning collaboratively trains a global model across decentralized clients without sharing raw data. In the presence of intermittent client availability, heterogeneous computational capabilities, and non-IID data distributions, federated learning can suffer from issues such as objective inconsistency Wang et al. (2020) and client drift Shi et al. (2022), leading to degraded performance. As data and client heterogeneity increases, balancing efficient and stable global convergence across clients remains a challenge.

Prior work addresses these issues through aggregation adjustments Acar et al. (2021); Pathak & Wainwright (2020); Karimireddy et al. (2020); Li et al. (2020a,b); Malinovsky et al. (2023); Charles & Konečný (2020), momentum-based updates Das et al. (2022); Xu & Huang (2022); Khanduri et al. (2021), and Newton-like methods Li et al. (2019). However, convergence of these client and server-side update strategies often depends on carefully tuned hyperparameters. The performance of federated learning is sensitive to the choice of hyperparameters such as learning rates and regularization terms Kim et al. (2023); Acar et al. (2021); Li

et al. (2020b). The challenge of selecting appropriate hyperparameters is compounded as the hyperparameter search space scales combinatorially with the number of clients. Practical implementations often apply a uniform hyperparameter configuration across all clients to avoid searching the entire hyperparameter space; however, this approach often yields suboptimal performance due to data and computational heterogeneity. To address this, adaptive methods have been proposed, typically focusing on either client step size selection or server-side aggregations.

Server-side adaptive methods adjust learning rates and aggregation strategies to account for system heterogeneity. For example, SCAFFOLD Karimireddy et al. (2020) uses control variates to reduce client drift, while FedYogi, FedAdaGrad, and FedAdam Reddi et al. (2020) extend adaptive gradient methods to federated learning aggregation steps in order to stabilize noisy updates. FedExp Jhunjhunwala et al. (2023) applies exponential moving averages to smooth aggregation, and FedDyn Acar et al. (2021) introduces a dynamic regularizer to promote stability. However, these methods often rely on prior knowledge of client learning rates, limiting their ability to support client-specific and non-uniform step sizes. Moreover, these aggregation strategies introduce additional hyperparameters that affect the convergence and stability of the global model.

On the other hand, client-side adaptive methods focus on dynamically adjusting client updates to handle heterogeneity. FedProx Li et al. (2020b) adds a proximal term to limit client divergence, while MOON Li et al. (2021) modifies client objectives to better align local heterogeneity. Auto-tuning methods like Kim et al. (2023) adapt client learning rates based on local gradients. Although these methods improve client optimization, without a coordinated server aggregation, these client-side methods alone can be prone to issues in heterogeneous federated learning including client drift and degraded performance.

In this work, we introduce a fully adaptive federated learning method that dynamically tunes hyperparameters for *both* individual client updates and central server aggregation for efficient and stable convergence in heterogeneous settings. Our approach is based on a dynamical system model of federated learning Agarwal et al. (2025) and draws on principles from numerical simulation and circuit design to select learning rates as well as momentum parameters. Through this perspective, constructing updates for heterogeneous federated learning is effectively translated into designing an adaptive simulation engine. Specifically, by mapping the dynamical system representation to an equivalent circuit, we are able to adopt well-established circuit design and simulation principles to choose step-sizes and momentum terms.

The key advancement over Agarwal et al. (2025) lies in reformulating hyperparameter selection as a circuit design problem. While Agarwal et al. (2025) introduces the foundational circuit mapping, critical parameters such as time-step sizes and inductance values still require manual tuning. Moreover, allowing each client to adopt its own momentum and step-size parameters would cause the hyperparameter space to grow combinatorially with the number of clients, making conventional tuning approaches impractical. Adaptive FedECADO addresses this limitation by analytically deriving client-specific momentum parameters through the critical damping condition, and by employing local truncation error (LTE)-based adaptive time-stepping for both client and server updates. This results in dynamically adjusted, client-specific learning rates and momentum terms that naturally adapt to local gradient geometry, eliminating the need for problem-specific hyperparameter tuning.

First, our methodology selects client learning rates based on numerical integration principles to ensure accuracy. This allows each client to operate with independent, data-specific learning rates that adapt to the local gradient space. To address objective inconsistency, we employ an integration/extrapolation operator at the central server's aggregation step that aligns the client updates in synchronous timescale. Then, on the server side, we adapt learning rate and momentum parameters using insights from circuit design, to promote fast and stable convergence.

**Our main contribution** is a fully adaptive, end-to-end federated learning method for heterogeneous settings in which both client and server updates are dynamically controlled by a single global parameter. Crucially, this parameter emerges naturally from a circuit-based dynamical system formulation of federated learning, which provides a physically grounded abstraction for designing momentum values under client heterogeneity. By treating the federated learning trajectory as a simulation, we impose numerical stability and error-control principles from circuit simulation directly on the learning process—capabilities that are difficult to express or enforce within conventional discrete optimization frameworks.

As a result, the performance of our method is largely insensitive to the value of the global control parameter, eliminating the need for hyperparameter tuning and making the system practical to deploy at scale. We benchmark our approach across a wide range of federated learning scenarios with heterogeneous and non-IID data distributions, and demonstrate that our method achieves strong model performance without tuning, consistently outperforming prior adaptive approaches that depend on carefully selected hyperparameters. These results underscore the necessity of the circuit formulation for enabling principled, stable, and robust adaptation in heterogeneous federated learning.

## 2  Dynamical Model of Federated Learning

The challenges associated with heterogeneous federated learning (such as client drift and objective inconsistency) and hyperparameter tuning stem from discrete client updates that struggle to handle variability in data and client computation. Rather than addressing these issues directly by tuning the hyperparameters of the discrete update, we take a different approach by modeling federated learning as a continuous-time dynamical system.

The dynamical system model captures the evolution of both client and server states over continuous-time, with the goal of modeling the trajectory of the following global optimization problem:

$$\min_x \sum_i p_i f_i(x, \mathcal{D}_i), \tag{1}$$

where $x$ represents the global model parameters and $f_i$ represents the local loss function of a client, $i$, characterized by a local dataset $\mathcal{D}_i$. For simplicity, we denote $f_i(x) \equiv f_i(x, \mathcal{D}_i)$. The local objective functions are weighted by a scalar, $p_i$, representing the size of their local dataset Agarwal et al. (2025); Wang et al. (2020).

We adopt a dynamical system model for solving (1) from Agarwal et al. (2025) that represents the states of each client as $x_i(t)$ and the central agent model states as $x_c(t)$. To couple the client states with the central agent states, Agarwal et al. (2025) introduces a coupling flow vector, $I_{L_i}(t)$, that captures the *accumulated* difference between the local client state $x_i$ and the central state $x_c$ over the simulation time according to:

$$I_{L_i}(t) = L_i^{-1} \int_0^t (x_c(s) - x_i(s))\, ds, \tag{2}$$

with its time derivative given by:

$$\dot{I}_{L_i}(t) = L_i^{-1} (x_c(t) - x_i(t)). \tag{3}$$

The coupling vector, $I_{L_i}(t)$, is then integrated into the dynamical system modeling the continuous-time evolution of the client and central states:

$$\frac{d}{dt}x_c(t) + \sum_{i=1}^{|\mathcal{C}|} I_{L_i}(t) = 0 \tag{4}$$

$$L_i \dot{I}_{L_i}(t) = x_c(t) - x_i(t) \quad \forall i \in [1, |\mathcal{C}|] \tag{5}$$

$$-I_{L_i}(t) + \frac{d}{dt}x_i(t) + p_i \nabla f_i(x_i(t)) = 0 \quad \forall i \in [1, |\mathcal{C}|] \tag{6}$$

where $\mathcal{C}$ is the set of clients.

$I_{L_i}$ introduces a second-order dynamic that expedites the speed of convergence to steady-state Agarwal & Pileggi (2023). The hyperparameter $L_i$ represents a client-specific momentum term that controls how quickly the flow variable $I_{L_i}(t)$ responds to discrepancies between the global and local states; larger values of $L_i$ slow the rate of change, introducing smoother, more stable dynamics.

The dynamical system naturally converges toward a steady-state that achieves global consensus across all clients even when their data and compute capacities differ. This naturally accounts for heterogeneity in

clients. When the system reaches equilibrium, the vector $I_{L_i} \to 0$, signifying that $x_i \to x_c$ for all clients. The rate at which the system settles is determined by the hyperparameter $L_i$, which can be tuned to accelerate convergence.

This dynamical system model is inspired by an analog circuit shown in Appendix 6, where $I_L$ is mapped to an electronic component known as an inductor with an inductance of $L_i$.

## 2.1 Simulating the Dynamical System in Federated Learning Setting

From the perspective of the dynamical system model, training a federated model in a heterogeneous environment becomes a matter of accurately simulating (4)-(6) to its steady-state. Due to the distributed nature of federated learning, the simulation is partitioned across clients, where each computational node independently models the evolution of a client's state-variables and a central server aggregates these trajectories.

To distribute the simulation, we use the framework in Agarwal et al. (2025) to decouple the full set of coupled ordinary-differential equations (ODEs) (4)-(6) using an iterative Gauss-Seidel (G-S). This method separates each client's subproblem from the central agent by treating the coupling vector, $I_L^i$, as fixed from the previous iteration.

At the $k+1$ G-S iteration, active clients, denoted by the set $\mathcal{C}_a$, solve their local ODEs by simulating:

$$\dot{x}_i^{k+1}(t) + p_i \nabla f_i(x_i^{k+1}(t)) + I_{L_i}^k(t) = 0, \tag{7}$$

where $I_{L_i}^k(t)$ is treated as a constant coupling term from the previous iteration. Each client is simulated for a client-specific time-window $[0, T_i]$ which varies based on local computational capabilities.

Each active client then communicates the final state $x_i^{k+1}(T_i)$ to the central agent, which then updates the global state, $x_c^{k+1}, (t)$ and the coupling vector, $I_L^{i^{k+1}}(t)$, using the following coupled ODEs:

$$\dot{x}_c^{k+1}(t) = \sum_{i=1}^{n} I_{L_i}^{k+1}(t), \tag{8}$$

$$L_i \dot{I}_{L_i}^{k+1}(t) = x_c^{k+1} - I_{L_i}^{k+1}(t)\hat{G}_i^{th^{-1}} - x_i^{k+1}(t) + I_{L_i}^k(t)\hat{G}_i^{th^{-1}} \quad \forall i \in [1, |\mathcal{C}|]. \tag{9}$$

In this update, each client's state $x_i^{k+1}$ is assumed constant over the server's simulation.

The matrix $\hat{G}_i^{th}$ represents the *first-order sensitivity* of client $i$ to changes in the global state. Inspired by Thevenin impedances in circuit theory (which characterize how the current from a circuit component responds to changes in the voltage), $\hat{G}_i^{th}$ models how a client's state is expected to evolve in response to updates in the central agent state, $x_c$. This sensitivity allows the central agent to anticipate each client's response, leading to improved convergence as shown in Agarwal et al. (2025). The sensitivity matrix is:

$$\hat{G}_i^{th} = \frac{\partial I_{L_i}}{\partial x_i} = \frac{1}{\Delta t} + p_i \nabla^2 f_i(x_i), \tag{10}$$

where $\Delta t$ is the client step-size. The full derivation of $\hat{G}_i^{th}$ is provided in Agarwal & Pileggi (2023).

Computing the Hessian, $\nabla^2 f_i$, at every G-S iteration is computationally expensive. Instead, we approximate (10) with a *constant aggregate sensitivity* $\hat{G}_i^{th}$, computed by averaging the Hessian across a representative subset of local datapoints. This results in a client-sensitivity model as follows:

$$\hat{G}_i^{th} = \frac{1}{\Delta t} + p_i \bar{H}^i, \tag{11}$$

with $\bar{H}^i$ denoting the average Hessian over the sampled datapoints. In federated learning with non-IID data, each client's loss is shaped by its unique data distribution, leading to distinct local objectives. The Hessian $\nabla^2 f_i(x_i)$ captures the curvature of each client's loss and as a result, the sensitivity matrix, $\hat{G}_i^{th}$, directly reflects client data heterogeneity since non-IID distributions produce distinct sensitivity matrices across

clients. However, computing the full Hessian for large-scale models can become a significant computational bottleneck. To address this, our approach is also compatible with efficient Hessian approximations, such as Fisher information matrices Jhunjhunwala et al. (2024), as discussed in Appendix 19. Since the Hessian primarily serves as a weighting scheme across individual clients, these approximations are shown to have minimal impact on performance.

Using the continuous-time dynamical system model of federated learning, our goal is now to simulate the ODE for each client (7) and server aggregation (8)-(9) without relying on hyperparameters to achieve stable and efficient convergence under heterogeneity.

## 3   Designing Adaptive Simulation for Heterogeneous Federated Learning

We propose an end-to-end adaptive federated learning method that addresses the challenge of hyperparameter tuning in heterogeneous environments. Our approach is grounded in a continuous-time dynamical systems model (4)–(6), where both client updates and server aggregation are modeled as coupled ODEs. Unlike prior work Agarwal et al. (2025), which required carefully tuned client- and server-specific hyperparameters, we introduce a fully adaptive, momentum-driven federated learning method that eliminates the need for hyperparameter tuning. Our method achieves robust performance across a range of non-IID data distributions and varying client compute capacities. Specifically, we introduce:

1. **Client-side adaptation**: Each client independently simulates its local continuous-time dynamics defined in (7), using non-uniform, adaptive learning rates that are shaped by local gradient space and follow accuracy principles from numerical simulation.

2. **Server-side adaptation**: The server updates its state using client-specific momentum terms, which are chosen to expedite global convergence. Then, the server-side aggregation is performed by a provably stable numerical method with adaptive step-sizes to handle non-uniform client learning rates (to avoid objective inconsistency).

This framework enables stable and efficient training under heterogeneous conditions, and achieves fast convergence without requiring hand-tuned parameters.

### 3.1   Adaptive Client Updates

By viewing the evolution of client states as a continuous-time model, our goal is to accurately simulate the client ODE. The client ODE in (7) is simulated by marching forward in continuous time at discrete time-steps (or step-sizes) of $\Delta t_i$. The client state at the next time point, $t + \Delta t_i$, is defined by:

$$x_i(t + \Delta t) = x_i(t) + \int_t^{t+\Delta t_i} I_{L_i}^k(\tau) + \nabla f_i(x_i(\tau), \mathcal{D}_i), d\tau. \tag{12}$$

In general, the integral in (12) does not have a closed-form solution and instead is approximated using numerical integration methods. To minimize computational overhead, we opt for an explicit Forward-Euler (FE) integration method as follows:

$$x_i(t + \Delta t) = x_i(t) + \Delta t_i \left( I_{L_i}^k(t) + \nabla f_i(x_i(t), \mathcal{D}_i) \right), \tag{13}$$

where $\Delta t_i$ is the client-specific time step. While the update resembles gradient descent with learning rate $\Delta t_i$, it is derived from numerical simulation (further derivations are provided in Appendix 7). As a result, $\Delta t_i$ is not tuned for convergence to local minimum, but rather selected to ensure an accurate numerical approximation of the continuous-time dynamics in (12). This is guided by error bounds from numerical analysis.

The accuracy of a FE step is measured using the local truncation error (LTE), which captures the difference between the true ODE solution and its numerical approximation. The LTE for FE, denoted as $\varepsilon_{FE}$, is

estimated as:

$$\varepsilon_{FE} = \frac{\Delta t_i}{2}|I_{L_i}(t) + \nabla f_i(x_i(t), \mathcal{D}_i) - I_{L_i}(t - \Delta t) - \nabla f_i(x_i(t - \Delta t), \mathcal{D}_i)|. \tag{14}$$

The derivation is provided in Appendix 8. To ensure the discretized steps using FE accurately follow the client ODE in (7), we aim to select $\Delta t_i$ to maintain the LTE below a pre-defined tolerance $\gamma$:

$$\max(\varepsilon_{FE}) \leq \gamma. \tag{15}$$

This follows standard numerical simulation practices, where step sizes are selected based on local error control. To ensure efficient simulation, we use a backtracking line search to find the largest $\Delta t_i$ satisfying (15) (Appendix 11). However, Forward-Euler (and similarly gradient descent) updates are prone to divergence as numerical errors accumulate over time Cohen et al. (2024). We adopt prior work from circuit simulation Rohrer & Nosrati (1981) that determines the following $\Delta t$ that guarantees Forward-Euler stability:

$$\Delta t_{FE} \leq 2\frac{x_i^\top(\nabla f_i(x_i(t)) + I_{L_i})}{\dot{x}_i^\top(t)\dot{x}_i(t)}. \tag{16}$$

We use this bound as an initialization for adaptive step size selection, which is then refined through backtracking to improve LTE accuracy. This creates a client-specific adaptive step size that responds to the local geometry of each client's loss function.

In this work, we assume each active client performs local updates for a fixed wall-clock duration. Due to heterogeneity in compute capabilities, this leads to varying numbers of local epochs and learning rates. This corresponds to each client simulating its local ODE over a total time window of:

$$T_i = \sum_{j=1}^{e_i} \Delta t_i, \tag{17}$$

where $e_i$ is the number of local epochs determined by the client's computational capacity. To accurately coordinate active client updates, clients report their simulation window $T_i$ and $x_i(T_i)$.

## 3.2 Adaptive Server-Side Aggregation with Momentum

At each communication round, the central server receives both the final simulated state, $x_i(T_i)$, and the simulation window, $T_i$, from each active client. These updates are aggregated according to the global ODE defined in (8)–(9). Our goal is to design a server-side aggregation that is insensitive to hyperparameters by developing an adaptive simulation that accurately tracks the central agent ODEs.

This is accomplished in two steps. First, we design the momentum parameters, $L_i$, to ensure fast convergence in continuous-time. Then, we develop an adaptive simulation engine that accurately follows the designed ODE for stable convergence to the steady-state.

### 3.2.1 Selecting Momentum Parameters

The central agent dynamics, defined in equations (8)–(9), follow a linear ODE where the momentum parameter, $L_i$, plays a key role in both stability and convergence. This reflects a well-known challenge in momentum-based optimization (e.g., Nesterov's acceleration), where performance is sensitive to the choice of momentum-related hyperparameters. From a dynamical systems perspective, $L_i$ functions as *damping coefficients* that influences the convergence of the global state, $x_c(t)$, by shaping the eigenvalues of the system's linearized dynamics.

These systems can be designed to fall into one of three damping regimes as illustrated in Figure 1:

1. **Over-damped**: The system converges without oscillation and overshooting but more slowly, as the damping suppresses responsiveness.

2. **Under-damped**: The system converges with oscillatory behavior.

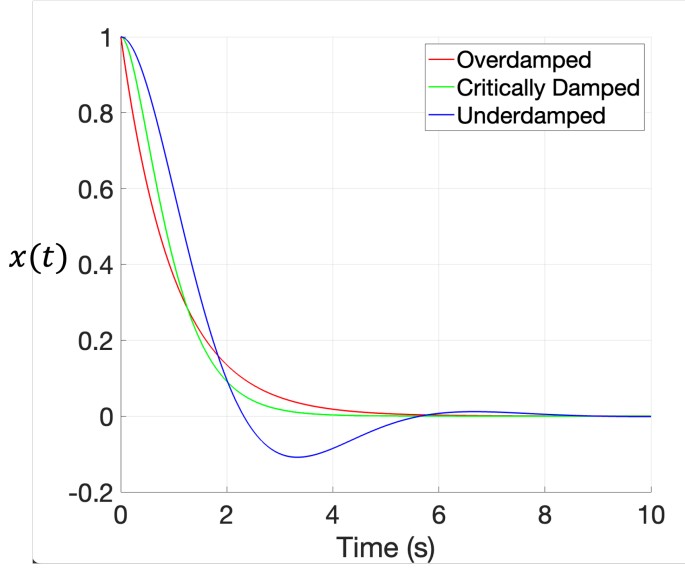

Figure 1: Simulating a single-variable second-order ODE, $\ddot{x}(t) + L\dot{x} + 10x(t) = 0$, we vary the parameter $L$ to achieve overdamped, critically damped, and underdamped systems.

3. **Critically damped**: The system converges quickly to steady-state without oscillation.

Our goal is to design the central agent ODE to be *critically damped* for fast convergence to steady-state without oscillations. Optimally selecting $L_i$ to achieve this requires controlling the eigenvalues of the system to lie at the boundary between overdamping and underdamping.

Mathematically, this can be achieved by optimizing:

$$\max_L inf \left( Re(-eig(\begin{bmatrix} C, 0 \\ 0, L \end{bmatrix}^{-1} G)) \right) \tag{18}$$

$$s.t. \quad Im(-eig(\begin{bmatrix} C, 0 \\ 0, L \end{bmatrix}^{-1} G)) = 0 \tag{19}$$

where $eig(\cdot)$ is the eigenvalue and $Re(\cdot)$ and $Im(\cdot)$ capture the real and imaginary components. $G$ is the collective client sensitivities. The full derivation is in Appendix 10 Solving (18) is computationally challenging, as eigenvalue computation is impractical with a large number of clients. Rather, we develop an approximate solution that uses the structure of federated learning to select a momentum parameter, $L_i$, that puts the system close to critical damping. To develop an approximate solution, we map the dynamical system to an analog circuit in Appendix 6 where the momentum parameter translates into selecting an inductance. This enables us to use tools from circuit theory to design the inductance to achieve critical damping.

Our approach is inspired by Thevenin impedances from circuits, which is used to simplify the behavior of complex interconnected components. By treating each client as a branch in a larger system connected through a central aggregator, we can apply similar techniques to analyze how each client's flow vector, $I_{L_i}$, (modeled by inductor current) responds to the shared central state $x_c$. To motivate this analysis, we first examine the dynamics of a single client's coupling variable $I_{L_i}$ from the central agent equation (8):

$$I_{L_i}(t) = \dot{x}_c(t) - \sum_{j \neq i} I_{L_j}(t). \tag{20}$$

This reveals two sources of coupling for client $i$: a direct coupling to all other clients through $\sum_{j \neq i} I_{L_j}(t)$, and a coupling to the central agent dynamics through $\dot{x}_c(t)$. While the cross-client term $\sum_{j \neq i} I_{L_j}(t)$ appears

directly in the equation, the underlying mechanism is indirect — each client first updates the central agent state $x_c$, which then propagates to influence other clients in subsequent aggregation steps. The central agent therefore acts as the intermediary through which all inter-client interactions are mediated.

To quantify the relative strength of these two couplings, we study the sensitivity of $I_{L_i}$ with respect to the central state $x_c$:

$$\frac{\partial}{\partial x_c} I_{L_i}(t) = \frac{d}{dx_c} \dot{x}_c(t) - \sum_{j \neq i} \frac{\partial}{\partial x_c} I_{L_j}(t). \tag{21}$$

This sensitivity combines:

1. The influence of $I_{L_i}$ on $\dot{x}_c$ (i.e., $\frac{d}{dx_c}\dot{x}_c(t)$), which is the direct path through which client $i$ influences the central state, and

2. The indirect influence of $I_{L_i}$ on all other clients' inductor currents $\frac{\partial}{\partial x_c}I_{L_j}(t)$ ($j \neq i$), mediated through their shared connection to $x_c$.

For federated learning, the dominant contribution to this sensitivity comes from changes in the central agent — the magnitude of the sensitivity of $I_{L_i}$ to perturbations in $x_c$ dominates the magnitude of its sensitivity to perturbations in other clients' flow variables $I_{L_j}$ directly (i.e., $\left|\frac{\partial I_{L_i}}{\partial x_c}\right| \gg \left|\frac{\partial I_{L_i}}{\partial I_{L_j}}\right|, \forall j \neq i$). This is numerically demonstrated in Appendix 14.

This approximation is justified by how federated learning operates: clients do not communicate directly with one another but only through the central server. Updates from a client affect the global model, which then influences other clients in future rounds. Thus, at each step, the most immediate and significant effect of a client's update is on the central agent, not other clients. This allows us to approximate the sensitivity of the system to $I_L^i$ using only the central agent's response.

This approximation effectively decouples the clients from one another, allowing each to be analyzed independently in relation to the central server. Specifically, in equation (9), each client "sees" only the dynamics of the central agent's capacitor, $\dot{x}_c(t)$, rather than interacting with other clients. As a result, we can model each client branch as an isolated second-order system as:

$$\dot{x}_c^{k+1}(t) = I_{L_i}^{k+1}(t), \tag{22}$$

$$L_i \dot{I}_{L_i}^{k+1}(t) = x_c^{k+1} - I_{L_i}^{k+1}(t)\hat{G}_i^{th^{-1}} - x_i^{k+1}(t) + I_{L_i}^k(t)\hat{G}_i^{th^{-1}}. \tag{23}$$

which can be mapped to a series RLC circuit (Figure 5) as derived in Appendix 14.

This decoupled view allows the momentum parameter $L_i$ to be designed independently for each client, without accounting for client interactions. By modeling each client-server connection in isolation, we can independently select $L_i$ to achieve critical damping. Using the RLC circuit model, we derive a closed-form expression for $L_i$ in Appendix 14 that ensures critical damping:

$$L_i = \frac{1}{4}\hat{G}_{th}^{i^{-2}}. \tag{24}$$

The result is a momentum term that guarantees fast, stable convergence for the central agent dynamics.

### 3.2.2 Adaptive Aggregation Updates

With momentum parameters set in (24), our next goal is to design an adaptive aggregation method for heterogeneous client updates. This involves building an adaptive simulation engine that (a) aligns client updates on a shared time scale to ensure objective consistency, and (b) uses a stable numerical integration scheme that adaptively selects the central agent's step size to accurately follow the ODE.

**Interpolating/Extrapolating Heterogeneous Updates to a Synchronous Timescale:** Due to client heterogeneity, active clients simulate their local ODEs for different simulation windows, $T_i$. Without coordinated aggregation, this mismatch can lead to objective inconsistency Wang et al. (2020). To align the client updates on a synchronous timescale, we uses a linear interpolation/extrapolation operator, $\Gamma(x_i(t), \tau)$, as shown in Agarwal et al. (2025), that evaluates each client's state at intermediate timepoints $\tau$:

$$\Gamma(x_i(t), \tau) = \frac{x_i(t_2) - x_i(t_1)}{t_2 - t_1}(\tau - t_1) + x_i(t_1), \tag{25}$$

This allows the central agent to evaluate its state over a synchronized window $\tau \in [t_0, t_0 + \max(T_i)]$, ensuring client updates are properly aligned. Without this, mismatched timescales would prevent convergence to a shared steady state, as shown in Agarwal et al. (2025). The central agent dynamics are then:

$$\dot{x}_c^{k+1}(\tau) = \sum_{i=1}^n I_{L_i}^{k+1}(\tau) \tag{26}$$

$$L_i \dot{I}_{L_i}^{k+1}(\tau) = x_c^{k+1}(\tau) - (I_{L_i}^{k+1}(\tau)G_i^{th^{-1}} + \Gamma(x_i^{k+1}(t), \tau) - I_{L_i}^k G_i^{th^{-1}}). \tag{27}$$

**Simulating Central Agent via Backward-Euler:** Next, the central agent ODE (26)-(27) is numerically simulated to solve for the central agent states. We propose using a numerically stable, Backward-Euler method that defines each aggregation step as:

$$x_c^{k+1}(\tau + \Delta t) = x_c^{k+1}(\tau) - \Delta t \sum_{i=1}^n I_{L_i}^{k+1}(\tau + \Delta t), \tag{28}$$

$$I_{L_i}^{k+1}(\tau + \Delta t) = I_{L_i}^{k+1}(\tau) + \frac{\Delta t}{L}\Big(x_c^{k+1}(\tau + \Delta t) - (I_{L_i}^{k+1}(\tau + \Delta t)G_i^{th^{-1}} + \Gamma(x_i^{k+1}(t), \tau + \Delta t) -$$
$$I_{L_i}^k(\tau + \Delta t)G_i^{th^{-1}})\Big). \tag{29}$$

The Backward-Euler method is unconditionally stable as derived in Appendix 9, meaning it converges to the system's steady-state (i.e., critical point of the objective) regardless of the step size $\Delta t$.

**Adaptive Time-Step Selection** Although the BE provides numerical robustness, our objective is to select a $\Delta t$ that accurately follows the central agent ODE designed for efficient convergence. We use an adaptive time-stepping scheme for the central agent in Agarwal et al. (2025) that selects the step size, $\Delta t$, based on the numerical accuracy of the BE rather than treating it as a hyperparameter.

The accuracy of the BE update is measured using the local truncation error (LTE). For the central agent dynamics in (26), the LTE is estimated as:

$$\varepsilon_{BE}^C = -\frac{\Delta t}{2}\left[\sum_{i=1}^n I_{L_i}^{k+1}(\tau) - \sum_{i=1}^n I_{L_i}^{k+1}(\tau + \Delta t)\right]. \tag{30}$$

The LTE for a BE integration of (27) is:

$$\varepsilon_{BE_i}^L = -\frac{\Delta t}{2L}\Big[(x_c^{k+1}(t) - I_{L_i}^{k+1}(t)\bar{G}_i^{-1} + x_i^{k+1}(t) - I_{L_i}^k(t)\bar{G}_i^{-1}) - (x_c^{k+1}(t + \Delta t) - I_{L_i}^{k+1}(t + \Delta t)\bar{G}_i^{-1} +$$
$$x_i^{k+1}(t + \Delta t) - I_{L_i}^k(t + \Delta t)\bar{G}_i^{-1})\Big]. \tag{31}$$

To ensure accurate simulation, we adaptively choose $\Delta t$ using a backtracking line search (Appendix 12) such that the maximum LTE across all components remains below a pre-defined tolerance $\gamma$:

$$\max|\varepsilon_{BE}| \le \gamma, \quad \varepsilon_{BE} = [\varepsilon_{BE}^C, \varepsilon_{BE}^L]. \tag{32}$$

### 3.3 Adaptive End-to-End Federated Learning

Our method is an adaptive, end-to-end federated learning algorithm designed to address both non-IID data and heterogeneous client computation. Computational heterogeneity is captured through an interpolation/extrapolation operator that aligns client updates, while data heterogeneity is modeled through the aggregate sensitivity matrix, which encodes differences in local curvature induced by non-IID data distributions.

A key element of our approach is the circuit perspective. By framing the system as an equivalent circuit, we can (i) visualize the topology of client–server interactions, and (ii) directly apply circuit design principles to construct the momentum term such that the overall system is critically damped. This perspective also enables us to leverage circuit simulation techniques to select stable step sizes, ensuring the discretized system remains passive and adapts naturally to both client and server gradient spaces.

This construction allows step sizes across clients and the server to be governed by a single global hyperparameter, $\gamma$, which controls the accuracy tolerance for numerical integration. While $\gamma$ is tunable, we show that the algorithm remains numerically stable across its range. The robustness of our method stems directly from the use of Backward-Euler integration. Unlike explicit methods, Backward-Euler is unconditionally stable and guarantees convergence regardless of the step size. As proven in Appendix 9, this property ensures that our algorithm converges to a stable optimum while tracking the continuous-time ODE dynamics designed by the momentum parameter $L_i$. Robustness is further reinforced by monitoring the local truncation error (LTE), as detailed in Appendix 12, which preserves the intended ODE behavior even under heterogeneous conditions.

The complete algorithm is provided in Appendix 13.

## 4 Experiments

Our methodology, called Adaptive FedECADO, is an adaptive federated learning method that achieves fast convergence via critical damping, while remaining robust to hyperparameter selection in heterogeneous settings. We study the effectiveness of our approach under heterogeneity in both data and computation. In our experiments, client data distributions follow a non-IID Dirichlet distribution, $|\mathcal{D}i| \sim \text{Dir16}(0.1)$, and each client performs a random number of local training epochs sampled from a uniform distribution according to $e_i \sim U[1, 50]$. In the following experiments, we employ a diagonal approximation of each client's Hessian to compute the aggregate sensitivity matrix $\hat{G}_{th}^i$. This Hessian approximation preserves relative weighting between client parameters and is computed once from a fixed mini-batch of representative points prior to the start of the simulation, and is reused across all subsequent rounds.

In these settings, Adaptive FedECADO achieves high model performance without requiring any manual tuning. In contrast, prior adaptive methods including SCAFFOLD Karimireddy et al. (2020), FedAdam Reddi et al. (2020), FedAdaGrad Reddi et al. (2020), delta-SGD Kim et al. (2023), FedCAda Zhou et al. (2025), and FAFED Wu et al. (2023) are sensitive to hyperparameter selections, leading to instability or poor convergence. Our results show that Adaptive FedECADO outperforms these baselines across multiple datasets and models, with faster convergence across a wide range of hyperparamter values.

Table 1: Percentage of useable models, defined as achieving over 80% of the highest accuracy, across all hyperparameter selections for CIFAR-10, CIFAR-100, and Sentiment-140 datasets.

| Dataset (Model) | Adaptive FedECADO | SCAFFOLD | FedAdam | FedAdaGrad | FedExp | deltaSGD | FedCAda | FAFED |
|---|---|---|---|---|---|---|---|---|
| CIFAR-10 (Resnet-18) | 100 | 42 | 15 | 15 | 0.05 | 50 | 75 | 90 |
| CIFAR-100 (Resnet-50) | 100 | 15 | 0.05 | 0.05 | 0 | 30 | 85 | 70 |
| Sentiment-140 (VGG-11) | 100 | 82 | 85 | 82 | 90 | 80 | 95 | 85 |

### 4.1 Robustness to Hyperparameter Selection

To evaluate robustness to hyperparameter selection, we perform a random search to identify the best-performing hyperparameters for each baseline method and for Adaptive FedECADO individually across multiple datasets and models, including CIFAR-10, CIFAR-100, and Sentiment-140 (shown in Appendix 16).

Figure 2 shows the final classification accuracies for each trial across the hyperparameters ranges reported in Appendix 15.

Adaptive FedECADO is shown to consistently achieve high accuracy with minimal variance, demonstrating strong robustness to its single hyperparameter, $\gamma$. In contrast, baseline methods show high sensitivity, performing well only under carefully tuned settings. Nearly all Adaptive FedECADO runs reach near-optimal performance, effectively removing the need for hyperparameter tuning. This is a combined result of (a) selecting momentum parameters that accelerate convergence and (b) adaptive numerical methods that ensure stable convergence. The contribution of each component is quantified through ablation studies in Appendix 21. We further highlight the advantages of our approach over Agarwal et al. (2025), demonstrating that Adaptive FedECADO matches the performance of a fully tuned FedECADO without requiring any manual tuning, eliminating the substantial effort of identifying client-specific momentum and step-size values that ensure stable and efficient convergence. A detailed breakdown of per-round wall-clock time across all baselines, per-step runtime proportions, and communication overhead are provided in Appendix 22. While Adaptive FedECADO introduces a slight increase in computational cost due to the dynamical system formulation and Backward-Euler steps, we argue that this overhead is justified by the method's improved robustness across hyperparameter settings, particularly as extensive hyperparameter tuning itself can incur greater computational cost in practice.

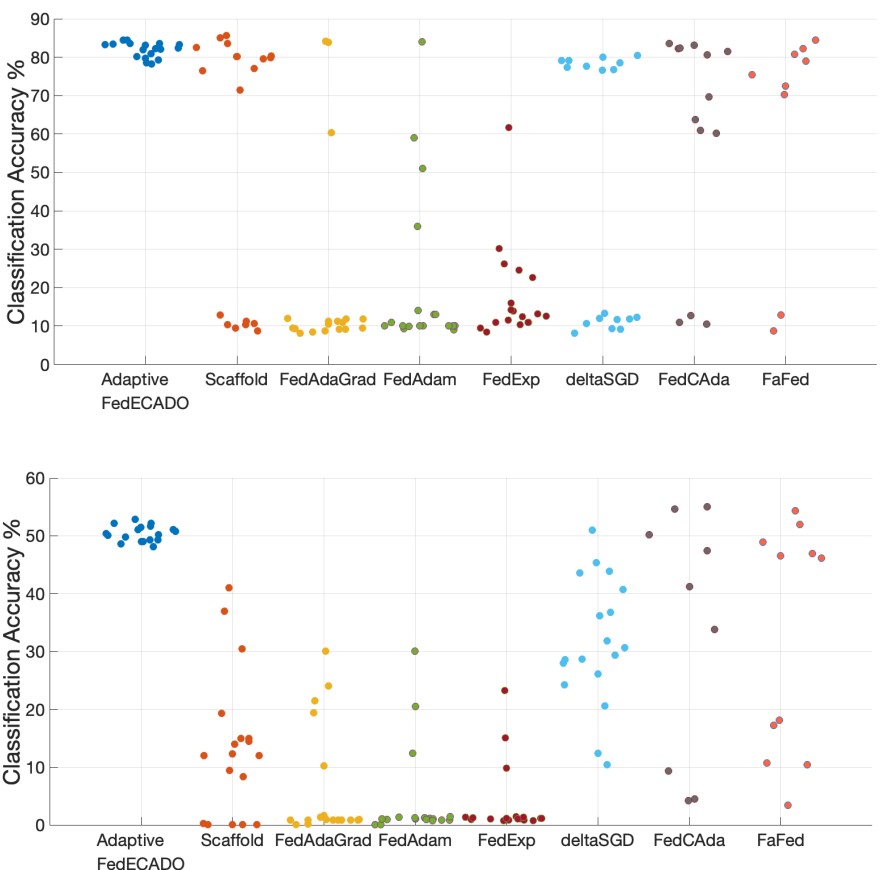

Figure 2: Classification accuracy of Adaptive FedECADO compared to baseline methods across a hyperparameter sweep (via random search for each method) under heterogeneous settings. Results are shown for (a) ResNet-18 on CIFAR-10, (b) ResNet-50 on CIFAR-100.

**Usable Rate of Federated Learning Methods:** A key advantage of a hyperparameter-free method is its ability to be deployed without additional engineering effort. To demonstrate practical deployability, we examine the number of usable models, defined as those achieving final accuracy above a deployment threshold

of 80% of the highest classification accuracy. Table 1 reports the percentage of hyperparameter configurations that produced usable models for each method.

In all experiments, Adaptive FedECADO achieves a usable rate of 100%, reliably producing high-performance models regardless of hyperparameter choices. In contrast, comparison methods often result in low-performance models that consume significant computational resources during training.

**Perturbing Hyperparameter Selections:** We also evaluate each method's sensitivity to hyperparameter perturbations by varying optimal values by 20% and measuring the impact on final accuracy (Appendix 17). Adaptive FedECADO remains stable and performant across all perturbations, showing robustness to its single hyperparameter, $\gamma$. In contrast, methods like FedAdam and FedAdaGrad exhibit significant performance drops, underscoring their dependence on precise tuning.

**Perturbing Hyperparameter Choices in Adaptive FedECADO**   We further evaluate the convergence behavior of Adaptive FedECADO under varying values of $\gamma$. Figure 7 shows training trajectories on the CIFAR-10 dataset using a ResNet-18 model across four orders of magnitude of $\gamma$, demonstrating robustness to its sole hyperparameter.

**Adding Schedulers to Client Updates**   Schedulers can be added to client updates; however, they introduce additional hyperparameters can improve stability at the cost of damping convergence speed. The impact of client-side schedulers to FedAdam and FedAdaGrad is shown in Appendix 18.

## 5   Conclusion

We introduce a fully adaptive federated learning method that eliminates the need for hyperparameter tuning in heterogeneous settings. By modeling federated learning as a dynamical system, our approach derives adaptive momentum and learning rates using principles from critical damping and adaptive simulation. The result is a momentum-based algorithm that achieves fast, stable convergence across non-IID data distributions and varying client compute, all controlled by a single global hyperparameter. We show that our method is highly robust to the choice of this hyperparameter. Compared to existing adaptive methods, Adaptive FedECADO delivers higher classification accuracy without tuning, whereas other methods are highly sensitive to hyperparameter selection. This makes our approach particularly well-suited for rapid prototyping by lowering the engineering overhead typically required to implement federated learning in heterogeneous environments. Future work will investigate Hessian approximations and more frequent curvature updates to better capture evolving client dynamics, as well as evaluation on larger-scale and more complex benchmarks where the client-local adaptivity of the method is expected to be most impactful.

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
