## Appendix

## 6    Equivalent Circuit Model of Federated Learning

Federated learning is a decentralized optimization framework in which a central server coordinates the training of a global model using gradient updates computed independently on distributed client devices. The global model parameters, $x_k$ at iteration $k$ are updated by gradient descent:

$$x_{k+1} = x_k - \alpha \sum_{i=1}^{N} \nabla f_i(x_k), \tag{33}$$

where $\nabla f_i(x_k)$ is the gradient of the local objective function for client $i$, and $\alpha$ is a global learning rate. As $\alpha \to 0$, this update can be represented as a time-discretized version of a continuous-time dynamical system:

$$\dot{x}(t) = - \sum_{i=1}^{N} \nabla f_i(x(t)), \tag{34}$$

where $\dot{x}(t)$ denotes the time derivative of the model parameters. While the continuous-time model in (34) offers useful insight into the dynamics of the optimization variables, it has a key limitation in the federated setting: all local clients use the global states, $x(t)$, to compute the local updates $\nabla f_i(x)$, however, in the federated setting, clients compute local updates using their own local states.

To better represent the structure of federated learning, Agarwal et al. (2023) introduced a continuous-time model that separates the global state from the individual client states using auxiliary flow vector, $I_{L_i}$. In this formulation, the global model, $x_c(t)$, is maintained by the central agent, and each client $i$ maintains its own local state denoted as $x_i(t)$. The interaction between a client and the server is captured through the auxiliary vector $I_{L_i}(t)$. This results in the following dynamical system:

$$\frac{d}{dt} x_c(t) + \sum_{i=1}^{|\mathcal{C}|} I_{L_i}(t) = 0, \tag{35}$$

$$L_i \dot{I_{L_i}}(t) = x_c(t) - x_i(t) \quad \forall i \in [1, |\mathcal{C}|], \tag{36}$$

$$-I_{L_i}(t) + \frac{d}{dt} x_i(t) + p_i \nabla f_i(x_i(t)) = 0 \quad \forall i \in [1, |\mathcal{C}|], \tag{37}$$

where $L_i$ is a scalar parameter that determines the coupling between client $i$ and the central agent.

In this work, we model the dynamical system in (35)-(37) as electrical circuits to leverage physical analysis and simulation techniques to design hyperparameters. Specifically, the dynamical system can be represented by the circuit in Figure 3 which mirrors the structure of the federated learning process. In this circuit, the central server states are modeled as voltages at a central node, $x_c(t)$, connected to node whose voltages represent the client state-variables, $x_i(t)$. These nodes are connected by flow variable $I_{L_i}(t)$ that represents the electrical current between the client and the central node.

In this circuit-based model, each client node is connected to a capacitor, which models the continuous-time dynamics of each client's state. A capacitor is an electrical circuit component that stores energy in an electric field and whose voltage changes based on the net current flowing into it. Mathematically, the current-voltage behavior of a capacitor is described by:

$$I_C = C\dot{V}, \tag{38}$$

where $I_C$ is the current, $C$ is the capacitance, and $\dot{V}$ is the rate of voltage change. In our analogy, the voltage across this capacitor corresponds to the local client states, $x_i(t)$. Each client node is connected to a voltage-controlled current source, which outputs a current equal to the local gradient $p_i \nabla f_i(x_i(t))$.

The client nodes are connected to the central node via inductors. An inductor is another basic circuit element that stores energy in a magnetic field and resists changes in current. It follows the current-voltage relation:

$$V_L = L\dot{I}, \tag{39}$$

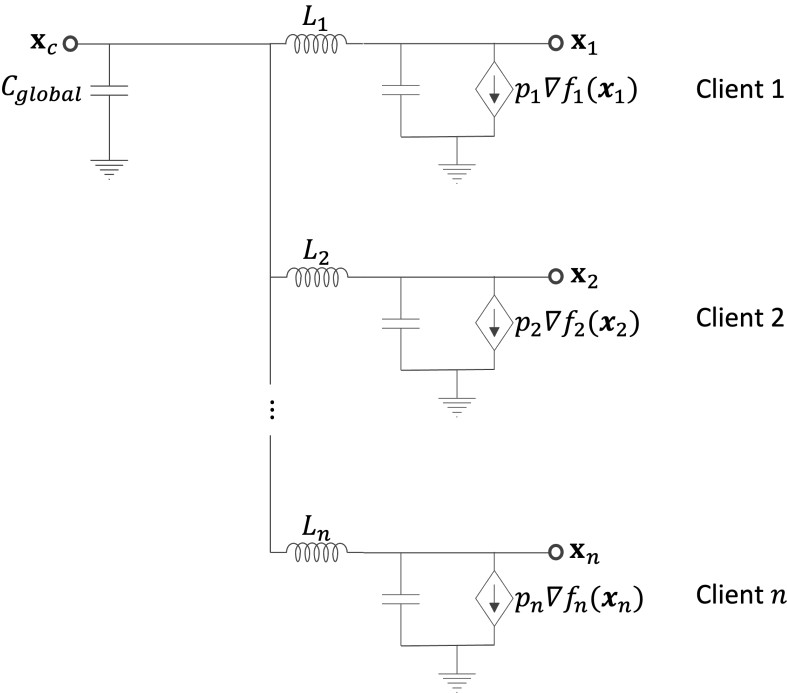

Figure 3: The dynamical system model of federated learning can be represented by an electrical circuit. In the circuit representation, each global state, $x_c$, is modeled by a node-voltage connected to multiple nodes whose voltage represents the client states, $x_i$. These nodes are connected by an inductor whose dynamics are modeled by (5).

where $V_L$ is the voltage across the inductor, $L$ is the inductance, and $I$ is the current. In our model, the voltage across the inductor is the difference between the global and client states, $x_c(t) - x_i(t)$, and the resulting current is $I_{L_i}(t)$. This implies that the inductor accumulates the difference between the global and local models over time, scaled by the inductance parameter $L_i$. Adding the inductors introduces a momentum-like effect into the client-server interaction.

Together, these components form a complete physical circuit, whose behavior is governed by Kirchhoff's Current Law (KCL). KCL is a fundamental principle in circuit theory which states that the total current entering a node must equal the total current leaving it.

Applying KCL at the central node yields the equation $\dot{x}_c(t) = -\sum_i I_{L_i}(t)$, representing the accumulation of all client currents into the global capacitor.

Applying KCL at each client node, the inductor current flowing into each client equals the sum of the capacitor current and the gradient-induced current. This equates to the client ODE $-I_{L_i}(t) + \frac{d}{dt}x_i(t) + p_i \nabla f_i(x_i(t)) = 0$.

By modeling the federated learning system using these physical components, we obtain a circuit-based representation that is structurally identical to the federated learning process. This perspective enables us to derive adaptive step sizes and momentum terms directly from physical principles, rather than relying on empirical tuning or heuristic modifications.

## 6.1 Modeling the Central Agent Aggregation as a Circuit

The continuous-time dynamics at aggregation round $k+1$, which can be described by the following system of equations:

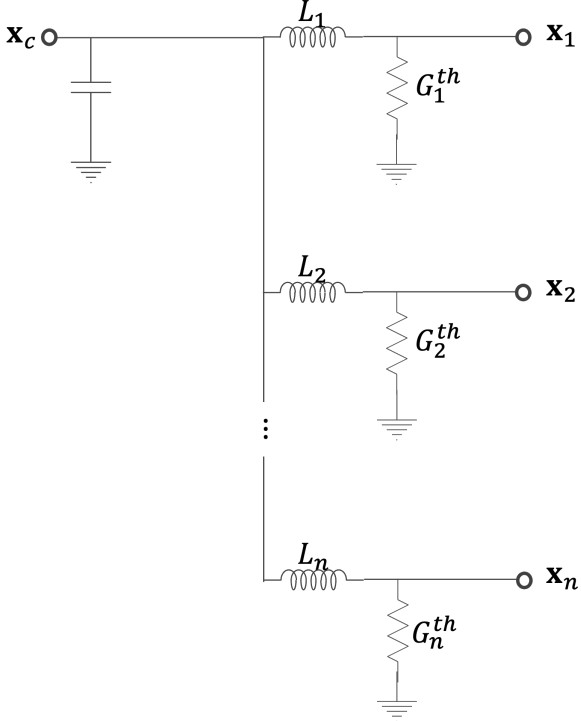

Figure 4: Equivalent circuit represented of the aggregation step for central agents in (8)-(9)

$$\dot{x}_c^{k+1}(t) = \sum_{i=1}^{n} I_{L_i}^{k+1}(t), \tag{40}$$

$$L\dot{I_{L_i}}^{k+1}(t) = x_c^{k+1}(t) - \left( I_{L_i}^{k+1}(t)\hat{G}_i^{\mathrm{th}^{-1}} + x_i^{k+1}(t) - I_{L_i}^{k}(t)\hat{G}_i^{\mathrm{th}^{-1}} \right), \tag{41}$$

where $\hat{G}_i^{\mathrm{th}}$ represents a linear sensitivity of each client, and $L_i$ is the momentum term for each client branch. This set of linear ODEs can be modeled by an electrical circuit shown in Figure 4.

This circuit differs from the previous nonlinear circuit representation of Figure 3 in that the client models are linearized via a first-order approximation using the sensitivity matrix $\hat{G}_i^{\mathrm{th}}$. Physically, the sensitivity term, $\hat{G}_i^{\mathrm{th}}$, behaves like a **resistor** in the circuit, which relates voltage and current linearly through Ohm's Law. This linear sensitivity is derived using a circuit concept known as **Thevenin impedance**, which models the sensitivity of current of a circuit network with respect to the node-voltage as an impedance and is used to simplify the analysis of linear electrical networks.

### 6.1.1 Client Perspective via Thevenin Impedance

The linear circuit in Figure 4 presents a complex interconnected network where the clients and central server states are coupled. The coupled interactions makes it challenging to derive momentum term, $L_i$, that makes the entire linear circuit critically damped.

Instead, we study the behavior of each individual client during aggregation using a *Thevenin impedance* looking outward from a single client branch into the rest of the circuit. From the client's perspective, this impedance includes the contribution of the central agent capacitor as well as all the other clients' branches connected in parallel. In our experiments we find that this Thevenin impedance is dominated by the central agent's capacitor, due to its relatively large capacitance compared to the resistance and inductance in the client branches.

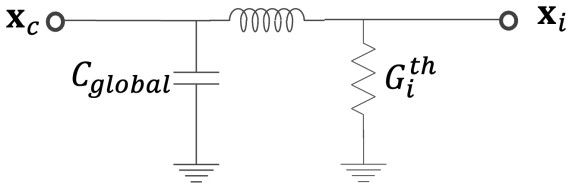

Figure 5: Each client branch in the aggregation circuit (Figure 4) can be reduced to a series RLC since the Thevenin impedance looking out of each client branch effectively looks like a capacitance for the central agent.

This observation allows us to approximate the Thevenin impedance looking out of the client branch as the central agent capacitor. As a result, each client branch looks like a **series RLC circuit**, consisting of:

- a resistor $R = \hat{G}_i^{\mathrm{th}^{-1}}$, representing the client's linearized sensitivity,

- an inductor $L_i$, representing the momentum term, and

- a capacitor $C_{global} = 1$, representing the central agent's state.

This reduced model is depicted in Figure 5. The series RLC circuit abstraction provides a highly interpretable local model for each client, enabling us to analyze convergence, damping, and oscillation behavior using well-known results from second-order systems. By modeling each client-server interaction as a damped oscillator, we can derive stability conditions and design update rules with predictable behavior, without relying on heuristic hyperparameter tuning.

### 6.2 Series RLC Behavior and Designing for Critical Damping

Each client-server connection in our federated learning framework can be modeled as a series RLC circuit. The behavior of this circuit is governed by a second-order differential equation that can be damped by tuning the inductance, $L_i$, as shown by Figure 1. Ideally, we want the system to be critically damped, meaning it returns to equilibrium as quickly as possible without oscillation. In the context of federated optimization, this corresponds to designing a momentum term for the aggregation step that achieves fast global convergence without overshooting.

To achieve critical damping in a series RLC circuit, we must design the inductance $L_i$ relative to the client sensitivity, $\hat{G}_i^{th}$, and the capacitance $C_{global} = 1$. We begin by recalling that the differential equation governing the series RLC circuit is:

$$L_i \ddot{q}(t) + R\dot{q}(t) + \frac{1}{C}q(t) = 0, \tag{42}$$

where $q(t)$ is the charge on the capacitor, $\dot{q}(t)$ is the capacitor current , $L_i$ is the inductance, $R = \hat{G}_i^{th^{-1}}$ is the resistance, and $C = 1$ is the capacitance. This equation has the same form as a damped second-order linear system and is derived in circuit literature Agarwal & Lang (2005). From this, we can derive a damping ratio $\zeta$, which determines whether the system is underdamped, critically damped, or overdamped, is given by:

$$\zeta = \frac{R}{2}\sqrt{\frac{C}{L_i}}. \tag{43}$$

Critical damping occurs when $\zeta = 1$, leading to the condition:

$$1 = \frac{R}{2}\sqrt{\frac{C}{L_i}}. \tag{44}$$

Solving for $L_i$, we square both sides:

$$1 = \frac{R^2 C}{4 L_i} \quad \Rightarrow \quad L_i = \frac{C R^2}{4}. \tag{45}$$

Thus, to achieve critical damping, the inductance must be set according to:

$$L_i = \frac{C R^2}{4} = \frac{1}{4} \hat{G}_i^{th^{-2}}. \tag{46}$$

This equation gives us a principled way to design the inductance (i.e., momentum term) based on the client sensitivity $\hat{G}_i^{th}$. Rather than relying on heuristics or manual tuning, we can compute the inductance directly to ensure optimal convergence behavior. In practice, this means that each client can be configured with a momentum that improves convergence speed in the global learning process.

## 7 Derivation of Client Update Rule from Numerical Simulation

The client update rule in (13) is obtained by applying a Forward-Euler (FE) numerical integration scheme to the coupled client ODE system. We walk through this derivation pointwise below.

**Step 1: The Coupled Client ODE System.** The client state $x_i(t)$ and inductor current $I_{L_i}(t)$ evolve according to the following coupled system of ODEs:

$$\dot{x}_i^{k+1}(t) = -p_i \nabla f_i(x_i^{k+1}(t)) - I_{L_i}^k(t), \tag{47}$$

$$L_i \dot{I}_{L_i}^{k+1}(t) = x_c^{k+1} - I_{L_i}^{k+1}(t) \hat{G}_i^{th^{-1}} - x_i^{k+1}(t) + I_{L_i}^k(t) \hat{G}_i^{th^{-1}}, \tag{48}$$

where $x_i(t)$ is the client model state, $x_c$ is the central server state, $I_{L_i}(t)$ is the inductor current representing the momentum state of client $i$, $p_i$ is a client-specific scaling parameter, $L_i$ is the inductance governing the momentum dynamics, and $\hat{G}_i^{th}$ is the client's aggregate sensitivity. The goal is to accurately simulate the evolution of this system forward in continuous time using discrete time-steps.

**Step 2: Integral Form Over One Time-Step.** The exact solution for $x_i$ over one time-step of size $\Delta t_i$ is given by:

$$x_i(t + \Delta t_i) = x_i(t) + \int_t^{t + \Delta t_i} \left( -p_i \nabla f_i(x_i(\tau), \mathcal{D}_i) - I_{L_i}^k(\tau) \right) d\tau. \tag{49}$$

In general, this integral does not admit a closed-form solution due to the nonlinearity of $\nabla f_i$.

**Step 3: Forward-Euler Approximation.** From numerical simulation, the integral in (49) can be approximated by using an explicit first-order Forward-Euler scheme, which evaluates the integrand at the current time point $t$ as:

$$\int_t^{t + \Delta t_i} \left( -p_i \nabla f_i(x_i(\tau), \mathcal{D}_i) - I_{L_i}^k(\tau) \right) d\tau \approx \Delta t_i \left( -p_i \nabla f_i(x_i(t), \mathcal{D}_i) - I_{L_i}^k(t) \right). \tag{50}$$

Substituting this approximation into (49) yields:

$$x_i(t + \Delta t_i) = x_i(t) - \Delta t_i \left( p_i \nabla f_i(x_i(t), \mathcal{D}_i) + I_{L_i}^k(t) \right). \tag{51}$$

**Step 4: Correspondence to Gradient Descent.** Equation (51) reveals that the Forward-Euler discretization of the client ODE takes the form of a momentum-augmented gradient descent step, where $\Delta t_i$ plays the role of the learning rate and $I_{L_i}^k(t)$ contributes a momentum correction term. Crucially, $\Delta t_i$ is not a manually specified learning rate but is instead determined by the numerical integration scheme: it is the time-step size required to accurately simulate the client ODE, and is set adaptively via local truncation error (LTE) control. This is the key distinction from standard gradient descent — the learning rate emerges from the simulation rather than being tuned by the practitioner.

## 8 Derivation of Local Truncation Error for Forward-Euler

To numerically solve the client ODE in (7), we apply the Forward Euler method with time step $\Delta t$. The discretized update rule is:

$$x_i(t + \Delta t) = x_i(t) - \Delta t \left( \nabla f(x_i(t)) + I_{L_i}(t) \right), \tag{52}$$

The local truncation error (LTE) measures the error introduced in a single step of the numerical method, defined as:

$$\varepsilon_{FE} = \frac{1}{\Delta t} \left[ x_i(t + \Delta t) - x_i(t) + \Delta t \left( \nabla f(x_i(t)) + I_{L_i}(t) \right) \right]. \tag{53}$$

To compute this, we expand $x_i(t + \Delta t)$ using a Taylor series about $t$:

$$x_i(t + \Delta t) = x_i(t) + \Delta t \dot{x}_i(t) + \frac{\Delta t^2}{2} \ddot{x}_i(t) + \mathcal{O}(\Delta t^3). \tag{54}$$

From the ODE (7), we know:

$$\dot{x}_i(t) = -\nabla f(x_i(t)) - I_{L_i}(t). \tag{55}$$

Substituting into the expansion:

$$x_i(t + \Delta t) = x_i(t) - \Delta t \left( \nabla f(x_i(t)) + I_{L_i}(t) \right) + \frac{\Delta t^2}{2} \ddot{x}_i(t) + \mathcal{O}(\Delta t^3). \tag{56}$$

Now, substituting into the expression for $\varepsilon_{FE}$:

$$
\begin{aligned}
\varepsilon_{FE_n} &= \frac{1}{\Delta t} \left[ -\Delta t \left( \nabla f(x_i(t)) + I_{L_i}(t) \right) + \frac{\Delta t^2}{2} \ddot{x}_i(t) + \mathcal{O}(\Delta t^3) + \Delta t \left( \nabla f(x_i(t)) + I_{L_i}(t) \right) \right] \\
&= \frac{1}{\Delta t} \left[ \frac{\Delta t^2}{2} \ddot{x}_i(t) + \mathcal{O}(\Delta t^3) \right] \\
&= \frac{\Delta t}{2} \ddot{x}_i(t) + \mathcal{O}(\Delta t^2).
\end{aligned}
\tag{57}
$$

In the expression, $\ddot{x}_i(t)$ can be evaluated as:

$$\ddot{x}_i(t) = I_{L_i}(t) + \nabla f_i(x_i(t)) - I_{L_i}(t - \Delta t) - \nabla f_i(x_i(t - \Delta t)). \tag{58}$$

Using this expression, the LTE for Forward-Euler is approximated by the first-order term as:

$$\varepsilon_{FE_n} = \frac{\Delta t}{2} \left[ I_{L_i}(t) + \nabla f_i(x_i(t)) - I_{L_i}(t - \Delta t) - \nabla f_i(x_i(t - \Delta t)) \right]. \tag{59}$$

## 9 Derivation for Backward-Euler Accuracy and Stability

The aggregation of client updates is modeled by a linear set of ODEs in (8)-(9). To numerically determine the central agent state, we discretize these ODEs using the Backward Euler method with a time step of $\Delta t$. The numerical update is:

$$x_c^{k+1}(\tau + \Delta t) = x_c^{k+1}(\tau) - \Delta t \sum_{i=1}^{n} I_{L_i}^{k+1}(\tau + \Delta t), \tag{60}$$

$$I_{L_i}^{k+1}(\tau + \Delta t) = I_{L_i}^{k+1}(\tau) + \frac{\Delta t}{L}\left(x_c^{k+1}(\tau + \Delta t) - (I_{L_i}^{k+1}(\tau + \Delta t)G_i^{\text{th}^{-1}}\right.$$
$$\left. + \Gamma(x_i^{k+1}(t), \tau + \Delta t) - I_{L_i}^k(\tau + \Delta t)G_i^{\text{th}^{-1}})\right). \tag{61}$$

To compute the local truncation error, we assume the previous value is exact and define the error for the first equation as:

$$\varepsilon_{BE}^C = \frac{1}{\Delta t}\left(x_c^{k+1}(t + \Delta t) - x_c^{k+1}(t) - \Delta t \sum_i I_{L_i}^{k+1}(t + \Delta t)\right), \tag{62}$$

and for the second equation:

$$\varepsilon_{BE}^L = \frac{1}{\Delta t}\left(L_i(I_{L_i}^{k+1}(t + \Delta t) - I_{L_i}^{k+1}(t))\right.$$
$$\left. - \Delta t\left[x_c^{k+1}(t + \Delta t) - \left(I_{L_i}^{k+1}(t + \Delta t)G_i^{\text{th}^{-1}} + \Gamma(x_i^{k+1}(t + \Delta t), \Delta t) - I_{L_i}^k G_i^{\text{th}^{-1}}\right)\right]\right). \tag{63}$$

Expanding the exact solutions using Taylor series:

$$x_c^{k+1}(t + \Delta t) = x_c^{k+1}(t) + \Delta t\dot{x}_c^{k+1}(t) + \frac{(\Delta t)^2}{2}\ddot{x}_c^{k+1}(t) + \mathcal{O}((\Delta t)^3), \tag{64}$$

$$I_{L_i}^{k+1}(t + \Delta t) = I_{L_i}^{k+1}(t) + \Delta t\dot{I}_{L_i}^{k+1}(t) + \frac{(\Delta t)^2}{2}\ddot{I}_{L_i}^{k+1}(t) + \mathcal{O}((\Delta t)^3). \tag{65}$$

Substituting into $\varepsilon_{BE}^C$:

$$\varepsilon_{BE}^C = \frac{1}{\Delta t}\left(\Delta t\dot{x}_c^{k+1}(t) + \frac{(\Delta t)^2}{2}\ddot{x}_c^{k+1}(t) - \Delta t \sum_i I_{L_i}^{k+1}(t) - (\Delta t)^2 \sum_i \dot{I}_{L_i}^{k+1}(t) + \mathcal{O}((\Delta t)^3)\right)$$
$$= \left(\dot{x}_c^{k+1}(t) - \sum_i I_{L_i}^{k+1}(t)\right) + \mathcal{O}(\Delta t). \tag{66}$$

Using the ODE definition, $\dot{x}_c^{k+1}(t) = \sum_i I_{L_i}^{k+1}(t)$, so:

$$\tau_n^{(x)} = \mathcal{O}(\Delta t). \tag{67}$$

A similar analysis for $\tau_n^{(I_{L_i})}$ shows that it is also $\mathcal{O}(\Delta t)$. Therefore, Backward Euler remains first-order accurate when applied to this linear system.

## 9.1 Numerical Stability of Backward Euler

The Backward-Euler integration is a numerically stable algorithm, which means for a linear system, we can guarantee convergence to the steady-state (or stationary point) regardless of the choice of $\Delta t$. To analyze property of numerical stability, we examine the linear system:

$$\dot{y}(t) = Ay(t), \tag{68}$$

where $A \succ 0$ is a linear system. The Backward Euler update is:

$$y_{n+1} = (I - \Delta t A)^{-1} y_n. \tag{69}$$

This scheme is stable if $eig(I - \Delta t A)^{-1} \leq 1$. For all $A \succ 0$, the value of $eig(I - \Delta t A)^{-1}$ is less than one. This implies that Backward Euler is **A-stable**: it remains stable for all step sizes $\Delta t > 0$, in the presence of stiff dynamics.

In the context of our aggregation ODE system, where the linear dynamics are governed by parameters $G_i^{\text{th}} \succ 0$ and $L_i > 0$, the eigenvalues of the linearized operator have negative real parts. Therefore, the Backward Euler method guarantees stability regardless of the time step $\Delta t$, making it a robust choice for simulating the dynamics of client-server interactions in heterogeneous and stiff federated learning systems.

## 10    Derivation of Damping Condition

Consider the linear set of ODEs representing the central agent aggregation (8)-(9). We abstract the linear ODEs to the form:

$$C\dot{x}(t) = Gx(t), \tag{70}$$

where $x(t) = \begin{bmatrix} x_c(t) \\ I_L(t) \end{bmatrix}$, $C = \begin{bmatrix} \mathcal{I}, 0 \\ 0, L \end{bmatrix}$ is a symmetric positive definite matrix representing capacitance and inductances. $G = \begin{bmatrix} 0, 0, ... \\ 0, \hat{G}_i^{th}, 0 \\ 0, 0, \ddots \end{bmatrix}$ is a positive semi-definite matrix representing client sensitivities.

Rewriting this system in standard form gives:

$$\dot{x}(t) = C^{-1}Gx(t). \tag{71}$$

The dynamics of this system are governed by the eigenvalues of the matrix $C^{-1}G$. Let $\lambda_i \in \mathbb{C}$ denote the eigenvalues of $C^{-1}G$. These eigenvalues determine the transient behavior of the system: real negative eigenvalues lead to exponential decay, while complex eigenvalues with nonzero imaginary parts introduce oscillations.

In control theory, a system is said to be **critically damped** when it returns to equilibrium as quickly as possible without oscillation. For a linear system, this corresponds to all eigenvalues of the system matrix being real and negative.

In the context of Equation (70), achieving critical damping involves two primary goals:

1. **Ensure all eigenvalues of $C^{-1}G$ are real and non-positive.** This removes imaginary components to avoid oscillations in the solution trajectories.

2. **Maximize the smallest (least negative) real eigenvalue of $C^{-1}G$.** This improves the convergence rate of the slowest mode and ensures the system returns to equilibrium quickly and uniformly.

Formally, if $\Lambda(C^{-1}G) = \{\lambda_1, \ldots, \lambda_d\} \subset \mathbb{R}$ denotes the eigenvalues of $C^{-1}G$, we aim to solve:

$$\max_{L \succ 0} \quad \inf_i \quad \text{Re}(\lambda_i(C^{-1}G)) \quad \text{subject to} \quad \text{Im}(\lambda_i(C^{-1}G)) = 0, \ \forall i. \tag{72}$$

This objective seeks the choice of inductance $L$ that spreads the eigenvalues of $C^{-1}G$ onto the negative real axis and pushes smallest $|\lambda_i|$ as far left as possible.

## 11 Adaptive Step Size Selection for Client Steps

The adaptive step-size selection procedure for an individual client is outlined in Algorithm 1. The algorithm is initialized with a time step $\Delta t$ based on the passivity conditions defined in Equation (16). A backtracking line search (line 4) is then employed to iteratively adjust the step size, ensuring that the local truncation error of the Forward-Euler integration for each client's ODE remains within the prescribed global tolerance $\gamma$ (line 3). The backtracking line-search damps the time-step according to the ratio, $0 < \frac{\gamma}{\max |\varepsilon_{FE}|} < 1$, which scales the step-size by how far the local truncation error is from the tolerance, $\gamma$.

---

**Algorithm 1** Adaptive Step Size Selection for Client Step

---

1: **Input:** $x_i^{k+1}(t)$, $I_{L_i}^k(t)$, $x_c^{k+1}(t)$, $\gamma > 0$
2: $\Delta t \leftarrow Equation(16)$
3: **while** $\max |\varepsilon_{FE}| \geq \gamma$ **do**
4: $\quad \Delta t = \frac{\gamma}{\max |\varepsilon_{FE}|} \Delta t$
5: $\quad \varepsilon_{FE} \leftarrow$ Equation (13)$(\Delta t)$
6: **end while**
7: **return** $\Delta t$

---

## 12 Adaptive Step Size Selection for Central Agent Aggregation

The adaptive step-size selection of the central agent aggregation is shown in Algorithm 2. This algorithm uses a backtracking line-search in line 4 to adaptive select a step-size that ensures the local truncation error of the Backward-Euler integration of the central agent ODEs are within a global tolerance, $\gamma$ (line 3).

---

**Algorithm 2** Adaptive Step Size Selection for Central Agent Aggregation

---

1: **Input:** $x_c^{k+1}(t)$, $I_{L_i}^k(t)$, $x_i^{k+1}(t)$, $\gamma > 0$, $\Delta t_k$
2: $\Delta t \leftarrow (\Delta t)_k$
3: **while** $\max |\varepsilon_{BE}| \geq \gamma$ **do**
4: $\quad \Delta t = \frac{\gamma}{\max |\varepsilon_{BE}|} \Delta t$
5: $\quad \varepsilon_{BE} \leftarrow$ (30), (31)
6: **end while**
7: **return** $\Delta t$

---

## 13 Adaptive Federated Learning Algorithm

The workflow for Adaptive FedECADO is shown in Algorithm 3. The input to the algorithm include each client's loss function and a global, scalar hyperparamter, $\gamma > 0$ that controls the local truncation errors for the central agent aggregation and all client simulations.

Prior to beginning the client communication, Adaptive FedECADO computes each client's sensitivity in line 5 as well as the corresponding momentum parameters in line 6. Then, a subset of active clients perform local updates using a Forward-Euler integration as shown in line 13, where the local step sizes adapt according to Algorithm 1. Each active client then communicates the final state vector and simulation window in line 14 to the central server. The aggregation of the client updates are then performed in line 18-19 according to a Backward-Euler integration of the central agent ODEs. The step-sizes for the central agent are determined using Algorithm 2, which is designed to control the local truncation error within a predefined tolerance, $\gamma$.

---

**Algorithm 3** Adaptive FedECADO Algorithm

---

**Input:** $\nabla f_i(\cdot), x(0), \gamma > 0$

1: $x_c \leftarrow x(0)$
2: $x_i \leftarrow x(0)$
3: $I_i^L \leftarrow 0$
4: $t \leftarrow 0$
5: Precompute $\bar{G}_i^{th} \; \forall i \in C$ via (11)
6: Precompute $L_i \; \forall i \in C$ via (24)
7: **do while** $\|\dot{x}_c\|^2 > 0$
8:     $x_c^k \leftarrow x_c^{k+1}$
9:     $x_i^k \leftarrow x_i^{k+1}$
10:    *Parallel Solve for active client states, $x_i^{k+1}(t + T_i) \forall i \in C_a$, by simulating:*
11:        **for $e_i$ epochs:**
12:            $\Delta t_i \leftarrow$ Algorithm 1
13:            $x_i^{k+1}(t + \Delta t_i) = x_i^{k+1}(t) - \Delta t_i \nabla f(x_i^{k+1}(t)) - \Delta t_i I_i^{L^k}(t)$
14:        Communicate $x_i(T_i)$ and $T_i$ to central server
15:    *Solve central agent aggregation by simulating:*
16:        **for $\tau \in [t, t + \max(T_i)]$**
17:            Select $\Delta t$ according to Algorithm 2
18:            Evaluate active client states at timepoint $\tau$: $\Gamma(x_i^{k+1}, \tau) \; \forall i \in C_a$
19:        *Solve for $x_c^{k+1}(\tau + \Delta t), I_i^{L^{k+1}}(\tau + \Delta t)$ according to (28)-(29)*
20:            $\tau = \tau + \Delta t$
21: Return $x_c$

---

# 14 Estimating Critical Damping

In Section 3.2.1, we develop an approximate solution to (18) that relies on the structure of the analog circuit model of federated learning. The approximation is developed by concluding that the sensitivity of $I_{L_i}$ with respect to the global state variable, $x_c$, is mainly attributed to the dynamics of the global state, $\frac{d}{dx_c}\dot{x}_c$.

To connect this to the sensitivity decomposition in Section 3.2.1, we recall that from equation (8), the coupling variable $I_{L_i}$ can be written as:

$$I_{L_i}(t) = \dot{x}_c(t) - \sum_{j \neq i} I_{L_j}(t). \tag{73}$$

This reveals two sources of coupling for client $i$: a coupling to the central agent dynamics through $\dot{x}_c(t)$, and a direct coupling to all other clients through $\sum_{j \neq i} I_{L_j}(t)$. The approximation in Section 3.2.1 is justified if the sensitivity of $I_{L_i}$ to perturbations in $x_c$ dominates its sensitivity to perturbations in the other clients' flow variables $I_{L_j}$ ($j \neq i$) directly. The negligible sensitivities $\frac{\partial I_{L_1}}{\partial I_{L_2}}$ and $\frac{\partial I_{L_1}}{\partial I_{L_3}}$ in Table 2 directly correspond to the cross-client terms $\sum_{j \neq i} \frac{\partial I_{L_j}}{\partial x_c}$ dropped in the sensitivity decomposition of Section 3.2.1. Their negligible magnitude empirically validates the decoupling assumption, confirming that the influence of other clients on $I_{L_i}$ is negligible relative to the central agent.

We can demonstrate this numerically by training a ResNet-18 model on the CIFAR-10 dataset through the federated setting across 3 clients. In this experiment, we create the linearized aggregation model in (8)-(9). Then we study the sensitivity of the flow variable of the first client, $I_{L_1}$, due to perturbations in the central agent state, $x_c$, and the flow variables of the other clients, $I_{L_2}$ and $I_{L_3}$. Table 2 demonstrates the normalized sensitivities across all state variables. We observe that the dominant sensitivity is due to $x_c$ as opposed to changes in the other clients' flow variables, with $\frac{\partial I_{L_1}}{\partial x_c} = 0.97 \gg \frac{\partial I_{L_1}}{\partial I_{L_2}} = 0.014$ and $\frac{\partial I_{L_1}}{\partial I_{L_3}} = 0.016$. This experimentally justifies that the dominant sensitivity term of any client is due to changes in the central agent state, and that cross-client coupling is negligible by comparison, thus allowing us to make the decoupling approximation in Section 3.2.1.

| Perturbation Variable | Normalized Sensitivity of $I_{L_1}$ to Perturbation Variable |
|:---:|:---:|
| $x_c$ | 0.97 |
| $I_{L_2}$ | 0.014 |
| $I_{L_3}$ | 0.016 |

Table 2: Sensitivity analysis of a single client flow vector $I_{L_1}$ in the central agent aggregation dynamics (8)–(9). We perturb the central agent state $x_c$ and the flow variables of clients 2 and 3, $I_{L_2}$ and $I_{L_3}$, and measure the resulting change in $I_{L_1}$. Sensitivities are computed as partial derivatives and normalized by the sum of all absolute sensitivities: $|\frac{\partial I_{L_1}}{\partial x_c}| + |\frac{\partial I_{L_1}}{\partial I_{L_2}}| + |\frac{\partial I_{L_1}}{\partial I_{L_3}}|$. The negligible sensitivities to $I_{L_2}$ and $I_{L_3}$ directly correspond to the cross-client terms dropped in the approximation of Section 3.2.1, empirically validating the decoupling assumption.

## 15 Varying Hyperparameters in Federated Learning Methods

We vary each federated learning method's hyperparameters to study their sensitivity to the model performance. The hyperparameters are randomly selected within the bounds listed in the following tables.

| Hyperparameter | Minimum Value | Maximum Value |
|:---|:---:|:---:|
| LTE tolerance $(\gamma)$ | 0 | $10^6$ |

Table 3: Hyperparameter ranges for Adaptive FedECADO

| Hyperparameter | Minimum Value | Maximum Value |
|:---|:---:|:---:|
| Global Step Size | 0 | 1 |
| Local Step Size | 0 | 1 |
| Initial control variate $(c)$ | 0 | 1 |

Table 4: Hyperparameter ranges for SCAFFOLD

| Hyperparameter | Minimum Value | Maximum Value |
|:---|:---:|:---:|
| Local Step Size | 0 | 1 |

Table 5: Hyperparameter ranges for FedExp

| Hyperparameter | Minimum Value | Maximum Value |
|:---|:---:|:---:|
| Global Step Size | 0 | 1 |
| Local Step Size | 0 | 1 |
| Momentum Factor $\beta_1$ | 0.9 | 1 |
| Momentum Factor $\beta_2$ | 0.9 | 1 |

Table 6: Hyperparameter ranges for FedAdam

| Hyperparameter | Minimum Value | Maximum Value |
|:---|:---:|:---:|
| Global Step Size | 0 | 1 |
| Local Step Size | 0 | 1 |
| Momentum Factor $\beta$ | 0.9 | 1 |

Table 7: Hyperparameter ranges for FedAdaGrad

## 16 Heterogeneous Federated Learning for Sentiment-140

We train a VGG-11 model on a Sentiment-140 dataset across 50 clients with an active client particition of 20%. We vary the hyperparameters of each adaptive federated learning method according to Appendix 15 and denote the final classification accuracies in Figure 6.

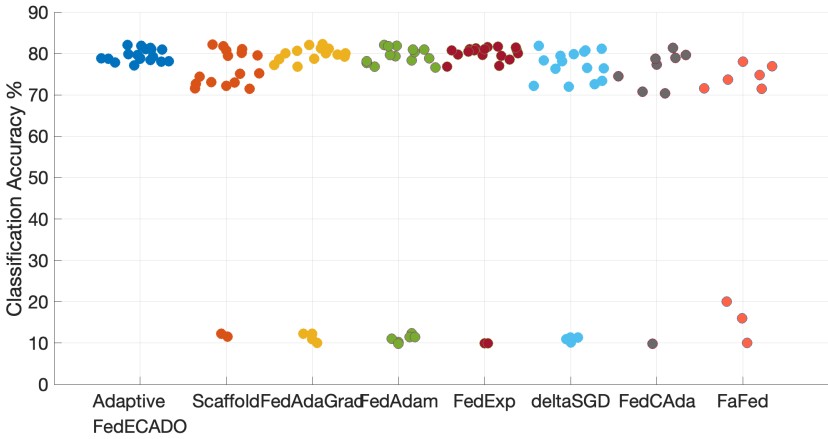

Figure 6: Hyperparameters for Adaptive FedECADO, SCAFFOLD, FedAdam, FedAdaGrad, FedExp and deltaSGD are swept during heterogeneous training for LSTM model for Sentiment140 dataset. The resulting classification accuracies of the random sweep are illustrated

| Dataset (Model) | Adaptive FedECADO | SCAFFOLD | FedAdam | FedAdaGrad | FedExp | deltaSGD | FedCAda | FaFed |
|---|---|---|---|---|---|---|---|---|
| CIFAR-10 (ResNet-18) | 81.9 (2.1) | 54.5 (34.6) | 21.2 (25.8) | 20.4 (21.3) | 27.7 (12.6) | 44.7 (34.7) | 60.1 (29.2) | 62.9 (29.9) |
| CIFAR-100 (ResNet-50) | 50.4 (1.37) | 13.4 (12.3) | 6.5 (1.2) | 4.31 (8.21) | 3.6 (6.2) | 31.6 (10.9) | 33.3 (21.6) | 32.2 (19.7) |
| Sentiment-140 (VGG-11) | 79.6 (1.49) | 73.32 (21.2) | 64.5 (29.3) | 62.4 (29.1) | 72.7 (22.9) | 63.4 (27.8) | 69.1 (22.5) | 54.7 (29.7) |

Table 8: Mean (Standard Deviation) of classification accuracies across all hyperparameter selections for training CIFAR-10, CIFAR-100, and Sentiment-140 datasets.

## 17 Perturbing Hyperparameter Selections

We study the effect of perturbing hyperparameters from their optimal values. The optimal hyperparemter selections for training CIFAR-10 on ResNet-18 model is determined from a hyperparameter sweep, with the final test accuracies of the sweep presented in Figure 2. We then perturb the hyperparameter values by 20% to see the effect on the model performance. As shown in Table 9, Adaptive FedECADO is highly insensitive to any perturbations where as many comparison methods can have large model performances.

The insensitivity of Adaptive FedECADO's single hyperparameter ($\gamma$) is especially highlighted by varying the value of $\gamma$ by four orders of magnitude in Figure 7, where we observe stable and efficient global convergence. The insensitivity to $\gamma$ is partly due to using a numerically stable Backward-Euler integration method for aggregating client updates.

| CIFAR-10 (Resnet-18) | Adaptive FedECADO | SCAFFOLD | FedAdam | FedAdaGrad | FedExp | deltaSGD | FedCAda | FaFed |
|---|---|---|---|---|---|---|---|---|
| Mean (Std) | 84.5 (1.3) | 85.1 (3.2) | 75.1 (10.2) | 70.4 (8.8) | 80.4 (6.4) | 82.2 (5.7) | 82.3 (6.4) | 80.9 (4.9) |

Table 9: Distribution of classification accuracies of federated learning methods trained under heterogeneous settings with hyperparameters randomly selected within 20% of the optimal values.

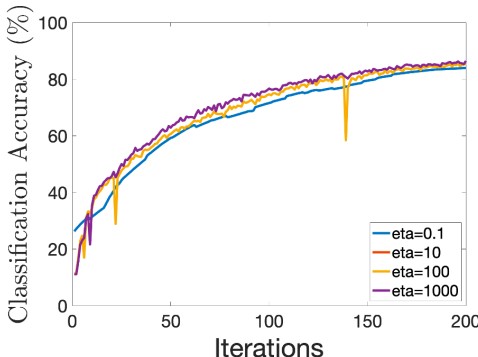

Figure 7: The hyperparameter of Adaptive FedECADO, $\gamma$ is swept across four order of magnitude in training CIFAR-10 dataset with ResNet-18 model under heterogeneous conditions. A wide range of $\gamma$ results in similar convergence plots.

## 18    Effect of Client Schedulers

In non-federated settings, schedulers are often used to adjust learning rates to achieve stable convergence. However, the challenge in including schedulers for each client in federated settings is that this creates non-uniform step sizes that requires proper coordination with the central agent aggregation step. Additionally, schedulers introduce their own hyperparameters that influence convergence rate, creating larger space of potential hyperparameter selections.

To demonstrate the challenge of introducing schedulers, we add exponential learning rate scheduler Li & Arora (2019) to client updates of FedAdaGrad and FedAdam aggregations and perform a random search of hyperparameter selections. The results, shown in Figure 8, indicate that schedulers results in fewer divergent cases, but slow the overall convergence for achieving a high-performing model. The hyperparameter for the exponential learning rate scheduler ($\gamma$) is randomly sampled from a uniform distribution, $U$ as $\gamma \sim U[0, 1]$.

## 19    Approximating Hessian Computation

Adaptive FedECADO requires computing the Hessian of each client via a smaller sample set before training to compute the optimal inductance parameters, $L_r$. However, computing the full Hessian may become computationally infeasible for large-scale models. In such circumstances, we can extend our framework to use Hessian approximations such as diagonal Fisher matrix Jhunjhunwala et al. (2024). We note that the main goal of the Hessian is not to accurately characterize the loss function, but is instead used as a relative weighting between clients. As a result, approximate hessian methods can still be applied with relatively high accuracy.

In this experiment, we study the the robustness of Adaptive FedECADO for training CIFAR-10 dataset on ResNet-18 model with random selections of LTE tolerance, $\eta$ using both full Hessian and approximate Hessian via Fischer matrix Jhunjhunwala et al. (2024). Figure 9 demonstrates that approximate Hessians offer similar performance to that of full Hessian. Our work provides a foundation for applying Hessian approximations in large models toward automated hyperparameter selection.

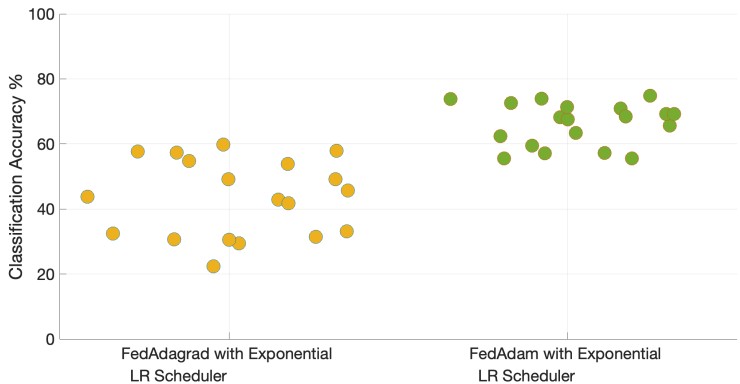

Figure 8: An exponential learning rate scheduler is added to the client updates with server aggregation performed by FedAdam and FedAdagrad Reddi et al. (2020). The hyperparameters of the server aggregation and client schedulers are randomly selected and the final classification accuracies of each run is shown.

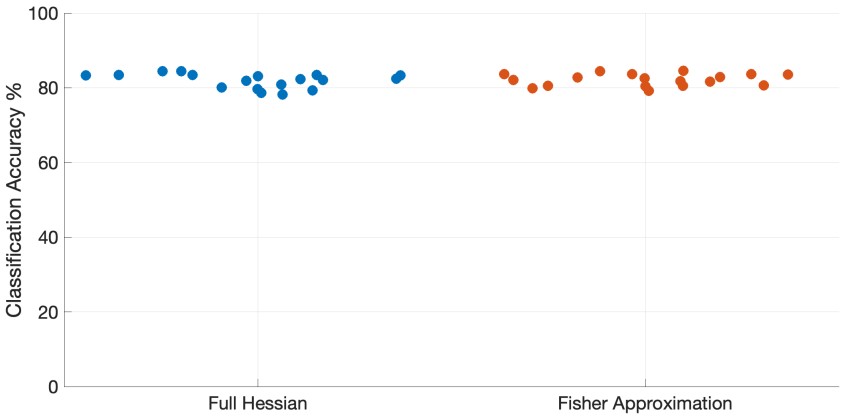

Figure 9: A ResNet-18 model is trained for CIFAR-10 dataset with Adaptive FedECADO using (a) full Hessian and (b) Fisher matrix approximation Jhunjhunwala et al. (2024) to compute the chord model.

## 20 Varying Configurations

To evaluate the generalizability of Adaptive FedECADO, we test its robustness under varying conditions of client participation and data heterogeneity. These experiments are designed to assess stability of our method under varying real-world settings. For each configuration, hyperparameters (listed in Appendix 15) are randomly selected for both baseline methods and Adaptive FedECADO, and we report the mean and standard deviation of classification accuracy across 20 independent runs (Tables 10–12). The results demonstrate the robustness of Adaptive FedECADO, evidenced by its consistently low variance across runs, while also achieving competitive or superior mean classification accuracy compared to baseline methods.

| $[C, k]$ | Adaptive FedECADO | SCAFFOLD | FedAdam | FedAdaGrad | FedExp | deltaSGD | FedCAda | FaFed |
|---|---|---|---|---|---|---|---|---|
| $C = 100, k = 0.3$ | 84.5 (1.5) | 53.6 (34.6) | 26.8 (25.8) | 27.8 (21.3) | 39.1 (32.6) | 51.8 (34.7) | 35.0 (29.2) | 53.5 (29.9) |
| $C = 100, k = 0.1$ | 81.1 (2.3) | 57.2 (39.9) | 21.0 (35.3) | 42.9 (28.7) | 29.7 (44.6) | 41.0 (28.5) | 46.8 (17.4) | 57.2 (19.2) |
| $C = 100, k = 0.05$ | 78.1 (2.1) | 50.4 (34.1) | 20.6 (39.3) | 21.2 (40.2) | 33.7 (31.2) | 46.8 (19.5) | 32.0 (24.1) | 45.3 (27.9) |
| $C = 20, k = 0.3$ | 87.4 (3.5) | 56.7 (23.0) | 33.2 (32.2) | 22.8 (31.3) | 23.5 (46.4) | 57.4 (15.7) | 31.3 (23.4) | 57.2 (14.2) |
| $C = 20, k = 0.1$ | 84.7 (5.4) | 47.1 (33.2) | 30.3 (40.2) | 26.0 (28.8) | 24.8 (32.8) | 56.5 (22.1) | 39.6 (26.7) | 50.4 (15.1) |

Table 10: Mean (Std) Classification Accuracy of CIFAR-10 (Resnet-18) trained under varying client participation settings (number of clients, $C$, participation ratio, $k$) with hyperparameters of federated learning methods randomly selected within operational range.

| $\delta$ | Adaptive FedECADO | SCAFFOLD | FedAdam | FedAdaGrad | FedExp | deltaSGD | FedCAda | FaFed |
|---|---|---|---|---|---|---|---|---|
| 0.05 | 80.9 (1.3) | 53.6 (24.7) | 22.1 (36.1) | 29.2 (37.8) | 26.6 (38.6) | 53.5 (14.2) | 53.8 (19.4) | 57.4 (12.7) |
| 0.1 | 84.2 (4.1) | 55.2 (23.2) | 25.0 (34.9) | 33.1 (26.6) | 26.5 (43.9) | 50.2 (15.8) | 60.6 (20.8) | 58.8 (14.2) |
| 0.5 | 86.6 (2.6) | 58.0 (20.8) | 37.5 (43.1) | 45.1 (26.3) | 33.1 (18.5) | 57.6 (21.0) | 62.5 (26.5) | 59.3 (13.3) |

Table 11: Classification Accuracy of CIFAR-10 (Resnet-18) trained under varying data distributions (Dir16($\delta$)) with hyperparameters of federated learning methods randomly selected within operational range.

| $\bar{e}$ | Adaptive FedECADO | SCAFFOLD | FedAdam | FedAdaGrad | FedExp | deltaSGD | FedCAda | FaFed |
|---|---|---|---|---|---|---|---|---|
| 80 | 87.4 (1.9) | 51.1 (32.1) | 39.8 (26.7) | 40.5 (18.1) | 40.2 (20.4) | 51.3 (27.7) | 51.2 (29.1) | 50.0 (24.2) |
| 30 | 83.1 (3.5) | 46.1 (36.8) | 24.9 (42.8) | 39.4 (43.6) | 35.4 (37.8) | 55.2 (13.5) | 40.9 (28.6) | 39.8 (26.1) |

Table 12: Mean (Std) Classification Accuracy of CIFAR-10 (Resnet-18) trained under varying distributions of number of local epochs ($e_i \sim U(0, \bar{e}]$) with hyperparameters of federated learning methods randomly selected within operational range.

Furthermore, we study the effect of different sampling distributions over the hyperparameter space. Figure 10 presents the performance of each optimizer under logarithmic sampling of server and client learning rates, drawn from LogUniform($10^{-4}, 10^0$). Consistent with the uniform sampling results, baseline federated learning methods remain highly sensitive to hyperparameter selection under logarithmic sampling, with a large proportion of configurations yielding unusable or divergent models. In contrast, Adaptive FedECADO consistently achieves near-optimal classification accuracy regardless of the sampled learning rate, as both client and server parameters are derived automatically from the local gradient geometry and dynamical system formulation rather than being drawn from a distribution.

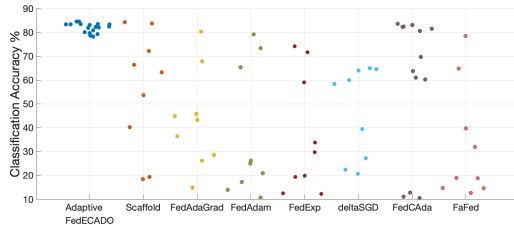

Figure 10: The classification accuracies of a ResNet-18 model trained for CIFAR-10 dataset is shown using learning rates sampled from a logarithmic distribution.

| Inductance Value, $L_i$ | Final Classification Accuracy |
|---|---|
| Adaptive client-specific $L_i$ from (24) | 85.1 |
| FedECADO with constant inductance $L = 0.1$ | 81.6 |
| FedECADO with constant inductance $L = 1$ | 84.7 |
| FedECADO with constant inductance $L = 10$ | 80.2 |

Table 13: Final classification accuracies for ResNet-18 on CIFAR-10 under varying inductance choices.

## 21 Comparison with FedECADO

Our work extends the circuit-theoretic formulation of federated learning in FedECADO Agarwal et al. (2025) by introducing a fully adaptive federated optimization scheme tailored to heterogeneous settings. In contrast to FedECADO, which requires careful manual tuning of momentum parameters and client step sizes to achieve good performance, our approach attains competitive—and often superior—accuracy without any manual hyperparameter tuning.

Specifically, our method adaptively controls both (i) the continuous-time dynamics through client-specific inductance parameters $L_i$, and (ii) the discrete-time simulation through coordinated choices of local client and central server time-steps. This joint adaptation ensures numerical stability of the discretized dynamics and reliable convergence to the steady state across heterogeneous clients.

While FedECADO relies on carefully selected inductance values to balance convergence speed and stability, our framework selects near-optimal inductance values, $L_i$, for fast convergence speed in continuous time without compromising stability of the underlying dynamics. To isolate the effect of the inductance parameters, we perform an ablation study in which all momentum parameters of FedECADO are varied by multiple orders of magnitude and is compared to our adaptive, client-specific inductance selection (computed via (24)).

As shown in Table 13-14, the performance of FedECADO is sensitive to the choice of inductance, with suboptimal values leading to noticeable degradation in final accuracy. In contrast, our adaptive inductance selection consistently achieves both faster convergence and the highest final accuracy, without requiring any momentum tuning.

Furthermore, the stability of our method is enforced by the proposed Backward-Euler numerical discretization scheme and LTE-controlled client step size selection. FedECADO is susceptible to divergence in local clients

| Inductance Value, $L_i$ | Final Classification Accuracy |
|---|---|
| Adaptive client-specific $L_i$ from (24) | 50.4 |
| FedECADO with constant inductance $L = 0.1$ | 44.3 |
| FedECADO with constant inductance $L = 1$ | 47.7 |
| FedECADO with constant inductance $L = 10$ | 50.6 |

Table 14: Final classification accuracies for ResNet-50 on CIFAR-100 under varying inductance choices.

| Local Client Step Size | Final Classification Accuracy |
|---|---|
| Adaptive client-specific step-size (LTE-controlled) | 50.4 |
| FedECADO with constant client step $\Delta t_i = 0.01$ | 43.56 |
| FedECADO with constant client step $\Delta t_i = 0.05$ | 45.23 |
| FedECADO with constant client step $\Delta t_i = 0.1$ | 46.68 |
| FedECADO with constant client step $\Delta t_i = 0.2$ | 51.00 |
| FedECADO with constant client step $\Delta t_i = 0.3$ | Divergence |

Table 15: Final classification accuracies for ResNet-50 on CIFAR-100 under varying local client step-size choices.

due to poorly selected step-sizes. Our methodology ensures that local clients adaptively select their step-sizes based on the gradient-space and is bounded by the global LTE value. As shown in Table 15

Furthermore, the stability of our method is enforced by the proposed Backward-Euler numerical discretization combined with LTE-controlled adaptive client step-size selection. In contrast, FedECADO is susceptible to divergence at local clients when step-sizes are poorly chosen, particularly in heterogeneous settings.

Our methodology ensures that each local client adaptively selects its step-size based on local gradient space and the resulting numerical error is globally bounded by an LTE threshold. To quantify the impact of client step-size selection, we compare our adaptive scheme against FedECADO with fixed client step-sizes spanning multiple orders of magnitude. The results for training ResNet-50 on CIFAR-100 are reported in Table 15.

Overall, our method uses insights from the circuit formalism derived in FedECADO but demonstrates a principled numerical discretization and circuit designed momentum values to yield stable and reliable convergence without the tuning requirements inherent to FedECADO.

## 22   Wallclock Runtime

The relative wall-clock times for all methods are presented in Table 16. While our method incurs slightly higher computational cost due to the dynamical system formulation and Backward-Euler steps, we believe that this overhead is compensated by the elimination of the hyperparameter tuning phase, which can be time-consuming and costly in federated environments.

| Method | Normalized Wallclock Time |
|---|---|
| Adaptive FedECADO | 1 |
| FedAdam | 0.94 |
| FedAdaGrad | 0.93 |
| SCAFFOLD | 0.96 |
| FedExp | 0.94 |
| deltaSGD | 0.97 |
| FedCAda | 0.98 |
| FAFED | 0.95 |

Table 16: Wallclock runtime (normalized to Adaptive FedECADO) for training a ResNet-18 model on the CIFAR-10 dataset over 100 epochs using different baseline methods.

**Per-Round Runtime Breakdown**: On the client side, the dominant cost is standard backpropagation, accounting for approximately 90–95% of client-side time and identical to all baselines. The remaining 5–10% is attributable to the local ODE time-stepping update. On the server side, the Backward-Euler integration step dominates at approximately 70–80% of server-side time; however, this step reduces to a pair of triangular solves against a pre-factorized LU decomposition computed once at the start of training, making it cheap in absolute terms. Server-side time-stepping accounts for approximately 15–20% of server-side time, with the LTE check comprising the remaining 5–10%.

**Communication Overhead**: Adaptive FedECADO introduces negligible additional communication cost relative to standard federated baselines. The only quantity communicated beyond gradients and model parameters is a single scalar, $\Delta t_i$, per client per round, representing the client's simulated time-period. This constitutes a near-minimal communication overhead regardless of model size, as the additional payload is independent of the model dimensionality $d$ and remains constant across rounds.

**Memory Requirements**: Regarding memory requirements, Adaptive FedECADO employs a diagonal approximation of the client sensitivity matrix $\hat{G}_{th}^i$, which reduces storage from $\mathcal{O}(d^2)$ for a full Hessian to $\mathcal{O}(d)$ per client. For the architectures considered in this work, the diagonal Hessian requires approximately 22.4MB (ResNet-18, $d = 11,689,512$), 48.8MB (ResNet-50, $d = 25,557,032$), and 253.9MB (VGG-11, $d = 132,863,336$) per client in 16-bit floating point precision. Since the LU factorization of a diagonal matrix is trivial — with $L = I$ and $U = \text{diag}(\hat{G}_{th}^i)$ — no additional storage beyond the diagonal itself is required, and the pre-computed factors are reused across all rounds at negligible per-round cost. The diagonal approximation is well-motivated in our setting, as $\hat{G}_{th}^i$ serves primarily to provide relative weighting between parameter dimensions rather than capturing full second-order interactions, and is consistent with widely used diagonal Hessian approximations in adaptive optimization (Goodfellow et al., 2016).

## 23 Using Default Hyperparameters

To further motivate the need for adaptive hyperparameter selection, we present an experiment training a ResNet-18 model on the CIFAR-10 dataset under heterogeneous data distributions. In this experiment, each client uses Adam with default hyperparameters and FedAvg aggregation with a server-side learning rate of 1, reflecting the setting a practitioner might naively adopt without problem-specific tuning. As shown in Figure 11, while the default configuration yields a functional model, it is far from optimal. Varying the client learning rates reveals that significantly better convergence properties are achievable. However, many alternative configurations produce unusable or divergent models, demonstrating that the default setting is neither reliably safe nor reliably optimal. This illustrates the core challenge: without knowledge of the data heterogeneity, model architecture, and client dynamics, there is no principled basis for selecting a default learning rate that generalizes across federated deployments, and practitioners risk poor convergence or divergence with no clear diagnostic signal.

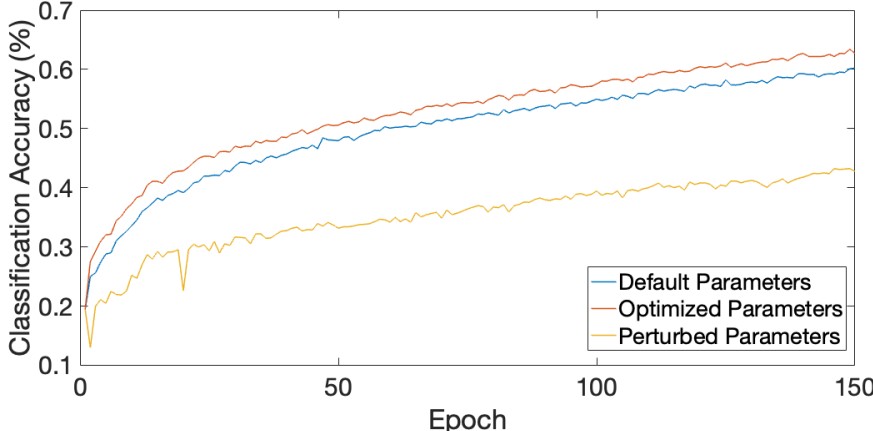

Figure 11: Classification accuracy over 150 epochs for ResNet-18 trained on CIFAR-10 under heterogeneous data distributions using FedAvg with a server-side learning rate of 1. Three client-side SGD configurations are shown: default learning rates, optimized client-specific learning rates, and perturbed configurations around the optimized values.