# OpenReview forum: "Adaptive Federated Learning via Dynamical System Model"
_TMLR — Accepted by TMLR_

### Review · Reviewer_jpCR · 2026-03-30

**Summary Of Contributions:**

The authors propose an adaptive modification to Agarwal et al.'s (2025) method for federated learning in the presence of client heterogeneity. Specifically, the authors propose a scheme for selecting the parameters $\Delta t_i$ and $L_i$ for each client. For $\Delta t_i$ (Section 3.1), the authors follow the stepsize-selection logic for the central agent (Section 3.2.2) previously proposed by Agarwal et al. (2025). For $L_i$, the authors use an approximation that decouples the clients from each other, allowing for an analytical expression for the critical dumping coefficient. The authors highlight the robustness of their approach on 3 datasets.

**Strengths**
1. The research question seems relevant.
2. The approach seems novel.

**Weaknesses**
1. The approximation in Section 3.2.1 does not seem sound. The authors leave out only one of several similar terms in Equation (8). From a theoretical point of view, this does not make sense since all of these terms play a similar role in aggregating the updates. From an empirical point of view, the authors refer to Appendix 14. However, in this appendix, they do not analyze the value they promise to analyze (i.e., they analyze $\partial I_i / \partial I_j$ and not $\sum_{j \neq i} \partial I_j / \partial x_c$).
2. Section 4 lacks details on the hyperparameter selection. Appendix 15 only reports the ranges for hyperparameters. However, the metric in Table 1 also crucially depends on the choice of distribution over these parameters.
3. Similarly, in Appendix 15, the ranges for learning rates are quite wide for the considered models and datasets, which might lead to inadequate comparison of the methods.
4. The paper does not discuss the computational and memory overheads of the method. First, even if the method's computational overhead is small, as the authors claim in Section 4.1, it is worth giving the reader a sense of its magnitude. Second, the method requires computing and storing the Hessian. Even if the cost of computing the Hessian can be reduced by using fewer points, the memory overhead does not vanish. In practice, often the memory overhead specifically prevents practitioners from implementing second-order methods. Thus, the authors should highlight this limitation and discuss ways to reduce the communication and memory overheads associated with the Hessian.
5. The paper lacks the proper attribution of the results. For instance, it might seem that Section 3.2.2 presents a novel algorithm for server-side step-size selection, but this algorithm was already proposed by Agarwal et al. (2025).
6. The paper is filled with typos. For instance, the linebreak in Equation (14) and similar linebreaks throughout the text hurt readability. The dot over $I_{L_i}$ is, for some reason. shifted to the right throughout the text. The notation for sensitivity changes randomly (e.g., compare Equations (9) and (11)).

**Audience:**

Yes

**Audience Explanation:**

I think the scope of the paper is rather narrow. However, the research question seems relevant for people interested in the framework of  Agarwal & Pileggi (2023).

**Broader Impact Concerns:**

I do not see the need for adding a Broader Impact Statement. Ethical implications of the work seem limited.

**Claims And Evidence:**

No

**Claims Explanation:**

My main concerns are Weaknesses 1, 2, and 3 outlined above.

1. Weakness 1 seems crucial for the conceptual correctness of the method.
2. Weaknesses 2 and 3 are crucial for the evaluation of the empirical gains of the method.

**Requested Changes:**

1. Please justify the approximation in Section 3.2.1.
2. Please give the details on the distribution of optimizers' hyperparameters. Provide some justification for the particular choice of distribution. Discuss the relevance to real-world practice of hyperparameter selection.
3. Please give a proper reference to  Agarwal et al. (2025) in Sections 2.1 and 3.2.2.
4. Please discuss the compute, memory, and communication overheads related to the method.
5. Please explain in more detail how to choose "representative points" for Hessian computation in Equation (11). Define $C$ in Equation (18).
5. Consider removing linebreaks in equations throughout the text. Fix notational inconsistencies.

---

> ### Author Response · Authors · 2026-04-27
>
> 1.  We thank the reviewer for their comments. To clarify the reasoning behind the approximation, we first explain why we study the sensitivity $\frac{\partial}{\partial x_c}I_{L_i}$.
> Our goal is to design $L_i$​ for each client independently, without solving the intractable eigenvalue optimization in (18)--(19). To do this, we examine the dynamics of a single client's coupling variable $I_{L_i}$ from (8), rewritten as :
> $$
> I_{L_i}(t) = \frac{d}{dt} x_c(t) - \sum_{j\neq i} I_{L_j}(t).
> $$
>
>  There are two sources of coupling for client i: a direct coupling to all other clients through $\sum_{j\neq i} I_{L_j}(t)$, and a coupling to the central agent dynamics through $\dot{x}_c(t)$.
>
>  To quantify the relative strength of these two couplings, we study the sensitivity of $I_{L_i}$ with respect to $x_c$, which from equation (8) is $ \frac{\partial}{\partial x_c} I_{L_i}(t) = \frac{d}{dx_c}\dot{x}_c(t) - \sum_{j\neq i} \frac{\partial}{\partial x_c}I_{L_j}(t) $.
>
>
>  Appendix 14 numerically demonstrates that
> $$
> \frac{d}{dx_c} \frac{d}{dt} x_c(t) \gg \sum_{j\neq i} \frac{\partial}{\partial x_c}I_{L_j}(t),
> $$
>  confirming that the central agent's contribution dominates. This indicates the influence of clients on one another is indirect — each client first updates the central agent state $x_c$, which then propagates to affect other clients in subsequent aggregation steps.
> From the circuit perspective, this is equivalent to the Thevenin impedance looking outward from any client branch being dominated by the central agent capacitor. This allows each client branch to be decoupled into an independent series RLC circuits, reducing the global eigenvalue problem to independent critical damping conditions, each yielding the closed-form expression $L_i = \frac{1}{4}\hat{G}_{th}^{i^{-2}}$.​
>
> We acknowledge that this motivation was not made sufficiently explicit in the manuscript, and we will add a clarifying remark connecting the sensitivity decomposition to the Thevenin impedance approximation in the revised manuscript.
>
> 2. We thank the reviewer for this observation. We clarify that the hyperparameter ranges in Appendix 15 are sampled uniformly. The ranges are intentionally broad because the optimal learning rate varies significantly with model architecture, dataset, and degree of client heterogeneity. We use a wide uniform range to reflect the diversity of configurations a practitioner might reasonably try. We acknowledge that tighter, problem-specific ranges could yield a more refined comparison, and is an interesting direction for future work. Nonetheless, we note that Adaptive FedECADO requires no hyperparameter tuning and its performance is unaffected by the choice of sampling range, making it well-suited for settings where such tuning is challenging.
>
> 3. Thank you for the suggestion. We have updated the references for those sections.
>
>
> 4. We appreciate the reviewer's request for greater transparency on computational costs. Wall-clock time comparisons across all baselines are reported in Appendix 22 (Table 22), and we add a pointer from the main text to improve discoverability. To further address the reviewer's concern, we provide the following breakdown of per-round costs and communication overhead, which is added to Appendix 22.
> Regarding communication overhead, Adaptive FedECADO introduces negligible additional cost relative to standard federated baselines. The only quantity communicated beyond gradients and model parameters is a single scalar $\Delta t_i$ per client per round, representing the client's simulated time-period. This overhead is independent of the model dimensionality $d$ and remains constant across rounds, constituting a near-minimal communication burden regardless of model size.
>
>
> 5. The representative points used to compute $\bar{H}^i$ in equation (11) are sampled as a random mini-batch from client 's local dataset once at the start of training and held fixed throughout. The Hessian is then averaged over this fixed mini-batch as
> $$
> \bar{H}^i = \frac{1}{\|\mathcal{B}_i\|} \sum_{d \in \mathcal{B}_i}  \nabla^2 f_i(x,d)
> $$
> where $\mathcal{B}_i$ is the sampled mini-batch. Computing $\bar{H}^i$ once and reusing it across rounds avoids the cost of repeated Hessian evaluations, and the LU factorization of the resulting system matrix is similarly computed once and saved. An interesting direction for future work is to explore the use of Hessian approximations as well as more frequent updates to $\bar{H}^i$ throughout training, which may better capture evolving local curvature at a manageable computational cost. We will add a clarifying remark on the selection and reuse of representative points to the revised manuscript.
>
>
> 6. Thank you for the suggestions. The equations are improved for readability in the revised manuscript.

---

> > ### Comment · Reviewer_jpCR · 2026-05-04
> >
> > Thank the authors for the response.
> >
> > 1\. Please look at Weakness 1. Appendix 14 does not address my concern.
> >
> > 2\. I appreciate that the authors clarified the scope of their experiment. However, I think that uniform sampling over such wide ranges is not a realistic baseline. First, the "mean" of the distribution is not realistic. If we assume completely cueless participants, they would probably use PyTorch defaults on average. If we assume that participants are more expert, they would probably use defaults for ImageNet training. Second, many practitioners use a logarithmic scale for choosing the learning rate. The current experiments do not follow this practice.
> >
> > 4\. Thanks for the clarification. However, again, I think that the memory requirements of the method should be highlighted.

---

> > > ### Author Response · Authors · 2026-05-12
> > >
> > > We thank the reviewer for this careful reading. We have highlighted the changes to the manuscript in the updated revision.
> > >
> > > 1.  We acknowledge that the connection between the claim in the main text and the analysis in Appendix 14 was not made explicit. In the revised manuscript, we demonstrate in Appendix 14 that:
> > >
> > > \begin{equation}
> > >     \left|\frac{\partial I_{L_1}}{\partial x_c}\right| = 0.97 \gg \left|\frac{\partial I_{L_1}}{\partial I_{L_2}}\right| = 0.014, \quad \left|\frac{\partial I_{L_1}}{\partial I_{L_3}}\right| = 0.016.
> > > \end{equation}
> > >
> > > By equation 8, which gives $I_{L_i}(t) = \frac{d}{dt} x_c (t) - \sum_{j\neq i} I_{L_j}(t)$,  the sensitivity of $I_{L_i}$ to $x_c$ is dominated by $ \frac{d}{dt} x_c (t)$. In the revised manuscript we have  updated the main text to frame the approximation directly in terms of  $\left|\frac{\partial I_{L_i}}{\partial x_c}\right| \gg \left|\frac{\partial I_{L_i}}{\partial I_{L_j}}\right|$, added a sentence making the equivalence to the cross-client sensitivity terms explicit via equation 8, and updated Appendix 14 to justify this assumption numerically.
> > >
> > >  2. We thank the reviewer for this suggestion. In the revised Appendix 20, we include a logarithmic sampling experiment to further demonstrate the robustness of Adaptive FedECADO compared to comparison methods. We observe similar behavior where many randomly selected client and server hyperparameter configurations produce unusable models in the comparison methods. In contrast, Adaptive FedECADO remains robust across a wide range of hyperparameter settings.
> > >
> > > We also wish to address the suggestion of using PyTorch defaults or ImageNet defaults as a baseline. While such defaults are reasonable starting points for centralized training, they do not transfer reliably to federated settings. We demonstrate this in Appendix 23 (Figure 11), where we train a ResNet-18 model on CIFAR-10 under heterogeneous data distributions using FedAvg with default SGD client learning rates and a server-side learning rate of 1. As shown in the figure, the default configuration yields suboptimal convergence compared to optimized client-specific learning rates. Furthermore, small perturbations away from the optimized configuration lead to meaningful degradation in performance, illustrating that the space around the optimum is narrow and difficult to navigate without extensive tuning. This sensitivity is compounded by the interaction between server and client learning rates. The combined hyperparameter space therefore grows combinatorially with the number of clients and it becomes challenging to borrow default hyperparameter selections from centralized settings. Adaptive FedECADO avoids the challenge of selecting optimal hyperparameters as both client and server parameters are derived automatically from the dynamical system formulation, requiring no manual selection at either level.
> > >
> > > 3. We have included a discussion on memory requirements in the revised Appendix 22

---

### Review · Reviewer_ZxA3 · 2026-04-13

**Summary Of Contributions:**

The paper proposes an adaptive federated learning framework that automatically adjusts learning rates for both clients and the server, removing the burden of manual hyperparameter tuning. It treats training as a continuous-time dynamical system, uses forward and backward numerical integration with local error control to set step sizes adaptively, and introduces critical damping to stabilize convergence.

**Audience:**

Yes

**Audience Explanation:**

This paper represents a genuine paradigm shift in how federated optimization is conceptualized. Translating the federated learning process into an equivalent circuit is exactly the kind of cross-disciplinary thinking that tends to open new research directions and is very valuable to the TMLR community. The presentation is polished, and the writing is of high quality. The authors have done an excellent job lowering the barrier to entry for what could have been an intimidating technical framework, especially if not from an electrical engineering background.

**Broader Impact Concerns:**

No impact concerns

**Claims And Evidence:**

Yes

**Claims Explanation:**

The paper is an exceptionally easy and enjoyable read. The narrative flows naturally from problem motivation through circuit analogy to algorithm design, making a technically dense contribution surprisingly accessible. The story the authors tell is coherent and persuasive: federated learning as a simulation problem, momentum as a damping design choice, and step sizes as numerical accuracy constraints. This reframing is genuinely elegant.

The practical motivation is strong as well. Hyperparameter tuning is a genuine bottleneck in federated deployment, and a principled, theory-backed path to eliminating it will be of direct interest to both researchers and practitioners. The breadth of experiments, across CIFAR-10, CIFAR-100, and Sentiment-140, with varying client counts, data heterogeneity levels, and local epoch distributions, gives readers enough context to assess where the method fits.

**Requested Changes:**

1. **(Critical to Acceptance)** The proposed method appears to be a direct extension of FedECADO in which the primary modification is replacing manually tuned hyperparameters with adaptive counterparts derived from numerical error control and critical damping. Given this, it is unclear why this contribution warrants a standalone paper rather than an ablation study or extension section within the original FedECADO work. The authors should articulate precisely what is architecturally or theoretically new beyond adaptive learning rates, and why that novelty is sufficient to constitute an independent contribution to the literature.

2. **(Critical to Acceptance)** On page 5, the authors state that equation (13) resembles gradient descent with learning rate $\Delta t_i$ but is derived from numerical simulation. This claim is made without any supporting derivation or intuitive walkthrough. Please add detailed pointwise steps on how $\Delta t_i$ is obtained. As a reader/practitioner, I am expecting the paper to provide me with steps on how to set up the numerical simulation, and how that simulation will provide the required $\Delta t_i$.

3. **(Critical to Acceptance)** The paper states that to solve the computationally intractable eigenvalue optimization in equations (18)-(19), the dynamical system is mapped to an analog circuit. However, it is never clarified whether this mapping produces a closed-form solution or merely reformulates the problem. If a closed-form expression does exist as suggested by equation (23), the paper should explicitly show how it is obtained from the circuit analogy and why it avoids the eigenvalue computation complexity flagged earlier. If practitioners are expected to simulate or construct a physical circuit to resolve this, that expectation must be stated and justified.

4. **(Critical to Acceptance)** Given that Adaptive FedECADO is positioned as a direct successor to FedECADO with adaptive hyperparameter selection, the most natural and informative comparison would be against a well-tuned FedECADO baseline. As a reader, I would like to see a systematic comparison against FedECADO operating near its optimal hyperparameter configuration. Without this, it is impossible to isolate how much of the performance gain comes from the adaptive mechanism itself versus the underlying dynamical system formulation that both methods share.

5. **(Strengthen the Work)** The approximation introduced on page 8 is a strong assumption whose validity is contingent on clients being effectively independent. In many real-world deployments of federated learning, clients are physically or operationally coupled subsystems. In such cases, the cross-client sensitivity terms omitted in equation (20) may be non-negligible, and the decoupled RLC approximation would no longer hold. The paper neither acknowledges this limitation nor analyzes how the method degrades when client interactions are present.

6. **(Strengthen the Work)** Equation (14) and equation (28) appear to contain formatting issues or missing terms that make them difficult to parse and verify. The authors should carefully review these expressions for completeness. More importantly, equations (39)-(40), which govern the central agent aggregation dynamics and are directly referenced throughout Sections 3.2 and 3.2.2, are relegated to the appendix. Since these equations are foundational to understanding the server-side update rule and the derivation of the momentum parameter, they should appear in the main paper rather than requiring the reader to navigate to supplementary material.

---

> ### Author Response · Authors · 2026-04-27
>
> 1. We appreciate the reviewer's question on the distinction between the two works. While FedECADO establishes the foundational mapping of federated learning to an equivalent circuit, several hyperparameters - including time-step sizes and inductance values - still require manual tuning, limiting its practical deployability. Adaptive FedECADO advances this foundation by leveraging the circuit analogy to reformulate hyperparameter selection as a circuit design problem. By treating each hyperparameter as a physical component, we can apply well-established circuit design principles - such as critical damping conditions - to derive adaptive, closed-form update rules that require no manual tuning. The core innovation lies in the derivation of $L_i$ via the critical damping condition and LTE-based adaptive time-stepping for both server and client steps. This enables client-specific learning rates and momentum terms that adapt to local gradient space and training dynamics, as well as server aggregation that adapts its local step sizes to synchronize heterogeneous client updates within a consistent simulated time-frame. We believe this constitutes an independent contribution beyond an ablation, as it introduces a principled design methodology that is transferable to future dynamical system-based federated learning methods. We highlight this in the introduction of the main text.
>
> 2. We thank the reviewer for this suggestion. The update rule in equation (13) is obtained by applying a Forward-Euler discretization to the client ODE in equations (7). The exact solution over one time-step is $x_i(t+\Delta t) = x_i(t) - \int_{t}^{t+\Delta t_i} I_{L_i}^k(\tau) + \nabla f_i(x_i(\tau), \mathcal{D}_i), d\tau.$,
>
> which does not admit a closed form due to the nonlinearity of $\nabla f$. Approximating the integral via Forward-Euler yields $x_i(t+\Delta t) = x_i(t) - \Delta t_i \left( I_{L_i}^k(t) + \nabla f_i(x_i(t), \mathcal{D}_i) \right)$, which takes the form of a momentum-augmented gradient descent step with learning rate $\Delta t_i$. Crucially, $\Delta t_i$ is not a manually specified hyperparameter but emerges directly from the numerical simulation via LTE-based adaptive step-size control, as detailed in the new Appendix 7 added in the revised manuscript.
>
> 3. We would like to clarify that the eigenvalue optimization in equations (18)--(19) is not solved directly; rather, it is reformulated as a circuit design problem in order to make the approximations in Section 3.2 more tractable. By modeling the linearized system as an equivalent circuit, we leverage circuit design techniques that represent each branch — encoding the hyperparameter $L_i$ alongside client training dynamics — as a physical component with well-understood design criteria. This reformulation yields a closed-form expression for the approximating the optimal $L_i$ via the critical damping condition, as given in equation (23), without requiring explicit eigenvalue computation. No physical circuit construction or simulation is expected of practitioners; the circuit analogy is a mathematical device that enables the closed-form result. We will add explicit clarifying language to this effect in the revised manuscript.
>
> 4. We agree that a direct comparison against a well-tuned FedECADO baseline most cleanly isolates the contribution of the adaptive mechanism, and such a comparison is provided in Appendix 22. Adaptive FedECADO matches the performance of a fully tuned FedECADO without requiring any manual tuning — achieving comparable convergence while eliminating the substantial effort required to identify client-specific momentum and step-size values that ensure stability and efficiency. We will add a pointer to this result from the main text in the revised manuscript.
>
> 5. We thank the reviewer for raising this point. We clarify that the decoupled RLC approximation is valid within the centralized federated learning setting assumed throughout this work. In this setting, the central server explicitly aggregates all client updates each round, which inherently accounts for cross-client interactions — the server-side aggregation step acts as the coupling mechanism between clients. The cross-client sensitivity terms in equation (20) are therefore not neglected in practice; rather, their effect is captured through the central agent dynamics. The approximation would only be insufficient in a fully decentralized setting where no central coordinator exists, which is outside the scope of this paper.
>
> 6. We thank the reviewer for catching these issues. In the revised manuscript, equations (14) and (28) have been carefully reviewed and corrected for readability. Additionally, we moved the references for central agent aggregation dynamics to the main text in the revised manuscript.

---

### Review · Reviewer_4v82 · 2026-04-20

**Summary Of Contributions:**

This paper addresses the bottleneck of hyperparameter tuning in heterogeneous federated environments through the following key contributions:

* Simplified Hyperparameter Management: Replaces multiple, complex tuning variables with a single global numerical error tolerance to dynamically govern both client-side learning rates and server-side aggregation.

* Continuous-Time Framework: Redefines the standard federated learning sequence as a continuous-time dynamical system, structurally mapped to an equivalent analog electrical circuit.

* Temporal Synchronization: Introduces a linear interpolation and extrapolation mechanism at the server level. This operator aligns client updates arriving at varying computational speeds onto a unified timeline, directly mitigating objective inconsistency and client drift.

* Analytic Momentum Tuning: Derives client-specific momentum values by treating network connections as decoupled RLC circuits. This mathematical formulation ensures a "critically damped" state, enabling swift convergence without oscillatory overshoot.

* Error-Bounded Step Sizes: Utilizes numerical simulation constraints to autonomously adjust learning speeds. Both clients and the server constrain their step sizes by bounding the Local Truncation Error (LTE) using Forward-Euler and unconditionally stable Backward-Euler integration, respectively.

**Audience:**

Yes

**Audience Explanation:**

The paper challenges the fundamental design paradigms of distributed optimization. While the issue of heterogeneous client constraints has been studied extensively, standard adaptive baselines (such as SCAFFOLD or FedAdam) still heavily rely on empirical tweaking and carefully scheduled tuning phases. This paper provides a refreshing, cross-disciplinary alternative by replacing these empirical heuristics with rigorous physical and numerical simulation principles. By proving that concepts like continuous-time modeling and critical damping can successfully govern distributed networks, the authors offer a theoretically grounded perspective that will strongly appeal to researchers interested in the intersection of dynamical systems, physics-informed machine learning, and federated architecture.

**Claims And Evidence:**

Yes

**Claims Explanation:**

The submission's claims are backed by a rigorous theoretical algorithms. Rather than relying on traditional heuristic evidence, the authors validate their framework through strict numerical constraints. By enforcing an LTE boundary via explicit Forward-Euler integration, the local devices naturally adapt their step sizes to the geometry of their specific loss landscapes. This self-regulating mechanism provides strong mathematical proof for their claim of reducing manual hyperparameter reliance.

Furthermore, the empirical evaluation design robustly supports the system's ability to handle severe network heterogeneity. To validate the effectiveness of the temporal alignment operator against client drift, the authors uniquely evaluate the system's "usable rate"—defined as the percentage of randomized configurations that yield a model achieving at least 80% of peak classification accuracy. By proving that the framework consistently produces viable models across a massive sweep of untuned setups, the evidence convincingly demonstrates its practical stability and resilience in unpredictable, real-world conditions.

**Requested Changes:**

The current evaluation methodology relies on a "random search" of hyperparameters to demonstrate a "usable rate" for the proposed method versus baselines like FedAdam and SCAFFOLD. Methods like FedAdam are specifically designed to perform optimally when correctly tuned, not under random selection.

The authors claim the method is "practical to deploy at scale" and brush oﬀ the computational overhead as "slight". However, computing the aggregate sensitivity matrix requires averaging the Hessian (or Fisher Information Matrix) over sampled datapoints, and the central server must compute unconditionally stable Backward-Euler integrations. It would be helpful it the authors could provide a concrete, tabular analysis of the wall-clock time and communication of their algorithm versus the baselines.

The empirical results rely entirely on outdated, small-scale benchmarks, the paper will become stronger if the authors could provide more complex and recent datasets and models.

---

> ### Author Response · Authors · 2026-04-27
>
> We would like to thank the reviewer for their comments and suggestions. Hope to address their comments below:
>
> ## Random Search
>
> We would like to clarify the intent of the random search experiment. Rather than assessing peak performance under optimal hyperparameter selection, the experiment measures the proportion of randomly sampled hyperparameter configurations that yield convergent, usable training runs, which we refer to as the usable rate. This metric aims to assess the challenge of performing exhaustive hyperparameter search in which many of the training runs often result in low performing models. The usable rate is a measure of how reliably a federated learning method can be deployed without extensive tuning, which is a practically relevant axis of comparison that complements peak-performance benchmarks.
>
>  Adaptive FedECADO is designed to alleviate precisely this burden by allowing each client to determine its own independent tuned hyperparameter based on local data statistics and learning dynamics. Methods such as FedAdam are indeed powerful when correctly tuned, but this tunability itself represents a deployment barrier in heterogeneous settings where a single global learning rate or momentum schedule may be ill-suited to the diversity of client distributions. The usable rate metric thus reflects a practically relevant measure of comparison that complements, rather than replaces, peak-performance benchmarks.
>
> ## Wall-Clock Time and Communication
>
> We appreciate the reviewer's request for greater transparency on computational costs. Wall-clock time comparisons across all baselines are reported in Appendix 22 (Table 22), and we will add a pointer from the main text. We have also provided a breakdown of per-round costs and communication overhead which will be added to Appendix 22. Regarding the computational overhead of the Hessian computation: the aggregate sensitivity matrix $\hat{G}_{th}^i$, is computed once at the start of training from a fixed mini-batch of representative points, and its LU factorization is pre-computed and saved. As a result, the per-round server-side cost reduces to a pair of triangular solves rather than a full matrix inversion making the overhead modest in practice as evidenced by Table 22. We also note that the method is designed to be general enough to accommodate future research into approximate Hessian methods, such as diagonal or low-rank estimates. In our work, the Hessian serves to provide relative weighting between clients rather than requiring exact curvature information. Replacing the exact Hessian with a cheaper approximation is therefore a natural direction for future work that would further reduce the computational burden without fundamentally altering the method's design principles.
>
> Regarding communication overhead, Adaptive FedECADO introduces negligible additional cost relative to standard federated baselines. The only quantity communicated beyond gradients and model parameters is a single scalar $\Delta t_i $per client per round, representing the client's simulated time-period. This overhead is independent of the model dimensionality $d$ and remains constant across rounds, constituting a near-minimal communication burden regardless of model size.
>
> ## Empirical Results:
>
> We thank the reviewer for this suggestion. The benchmarks were selected to provide a clear and interpretable proof-of-concept for Adaptive FedECADO, establishing that the method is viable before scaling to more complex settings. We agree that evaluation on larger-scale and more recent benchmarks is an important direction and have added a note to the conclusion reflecting this as a direction for future work.

---

### Decision · Action_Editor_mGdY · 2026-06-01

**Recommendation:** Accept as is

**Audience:**

Yes

**Audience Explanation:**

The paper deals with an important problem (the need for manual hyperparameter tuning) in federated learning. This should be of interest to people working in federated learning and distributed optimization. The proposed techniques and framework would also be of interest to people working in dynamical systems and physics-informed machine learning.

**Claims And Evidence:**

Yes

**Claims Explanation:**

The paper proposes Adaptive FedECADO, a modification to FedECADO (Agarwal et al. 2025) that allows the adaptive selection of parameters for different clients in the presence of client heterogeneity. The paper builds on the novel framework from the Agarwal et al. 2025 paper (casting the federated learning problem as a dynamical system, which is then mapped to an equivalent analog circuit) and extends the FedECADO approach it by proposing a scheme for selecting the step-size and a momentum parameter for each client. The reviewers agree that the paper provides good theoretical and empirical evidence for the efficacy of the proposed approach. The paper is well written and clear and there is agreement among the reviewers that manual hyperparameter tuning is an important issue in practical federated learning and that the contribution of the paper is worthwhile.

Some concerns were raised about the rigor in the approximation in page 8, which involves omitting cross-client sensitivity terms. The authors justified it by claiming that the coupling term to the central agent dynamics is much stronger than the contributions from other clients, making the cross-client terms negligible. Some empirical validation of this claim was provided.

Some concerns were also raised about the small scale of the benchmarks chosen. This was acknowledged by the authors and left as a future direction. This may limit the potential practical impact a bit, but does not significantly affect the value of the contribution.